# Long-range population dynamics of anatomically defined neocortical networks

**Jerry L Chen[1]\*[†‡], Fabian F Voigt[1,2][†], Mitra Javadzadeh[1], Roland Krueppel[1§], Fritjof Helmchen[1,2]\***

[1]Brain Research Institute, University of Zurich, Zurich, Switzerland; [2]Neuroscience Center Zurich, University of Zurich, ETH Zurich, Zurich, Switzerland

**Abstract** The coordination of activity across neocortical areas is essential for mammalian brain function. Understanding this process requires simultaneous functional measurements across the cortex. In order to dissociate direct cortico-cortical interactions from other sources of neuronal correlations, it is furthermore desirable to target cross-areal recordings to neuronal subpopulations that anatomically project between areas. Here, we combined anatomical tracers with a novel multi-area two-photon microscope to perform simultaneous calcium imaging across mouse primary (S1) and secondary (S2) somatosensory whisker cortex during texture discrimination behavior, specifically identifying feedforward and feedback neurons. We find that coordination of S1-S2 activity increases during motor behaviors such as goal-directed whisking and licking. This effect was not specific to identified feedforward and feedback neurons. However, these mutually projecting neurons especially participated in inter-areal coordination when motor behavior was paired with whisker-texture touches, suggesting that direct S1-S2 interactions are sensory-dependent. Our results demonstrate specific functional coordination of anatomically-identified projection neurons across sensory cortices.

\*For correspondence: jerry@chen-lab.org (JLC); helmchen@hifo.uzh.ch (FH)

[†]These authors contributed equally to this work

Present address: [‡]Department of Biology, Boston University, Boston, United States; [§]Federal Ministry of Education and Research, Bonn, Germany

**Competing interests:** The authors declare that no competing interests exist.

## Introduction

Sensory perception, fine voluntary motor control, and higher cognitive functions depend on neural dynamics in the mammalian neocortex, which itself relies on the exchange of information between cortical areas through both bottom-up (feedforward) and top-down (feedback) neuronal pathways across the cortical hierarchy (*Bressler and Menon, 2010*; *Buschman and Miller, 2007*). Cortico-cortical connections are formed between columnar microcircuits via long-range axons of pyramidal neurons in superficial layer 2/3 (L2/3) and deeper layer 5. A given cortical area typically establishes connectivity patterns not only with one particular area but with multiple target areas in a distributed and often reciprocal fashion (*Markov et al., 2013*; *Oh et al., 2014*; *Zingg et al., 2014*). Thus, in order to fully understand the cortical interactions underlying behavior, it is necessary to disentangle how neuronal subpopulations defined by both their functional properties and their specific anatomical projections contribute to local computation and long-range communication.

Such an understanding has been limited by the difficulty in measuring population activity across areas with sufficient spatial and temporal resolution. Present methods to study large-scale cortical dynamics either lack cellular resolution and sensitivity to low numbers of action potentials (e.g., human fMRI; *Hutchison et al., 2013*; or wide-field functional imaging in mice, *Ferezou et al., 2007*; *Lim et al., 2013*; *Minderer et al., 2012*) or they are restricted to poorly defined neuronal subsets as for extracellular recordings (*Melzer et al., 2006*). The main limitation for these recording approaches is the reliance on correlated activity patterns to infer information flow without the additional ability

**eLife digest** Behavior and cognition – the process of thought – emerge from computations that occur within vast networks of neurons in the brain. Within these networks, neurons may communicate with their neighbours in the same brain region as well as with distant counterparts in remote brain regions. Neuroscientists have studied these networks by measuring the activity of neurons within a single region or across the brain as a whole. However, it has not been possible to study long-distance communication between pairs of neurons in different brain regions. This has made it difficult to work out exactly what information brain regions exchange.

Chen, Voigt et al. now overcome these challenges by developing a new microscope system that allows researchers to measure the activity of individual neurons in different brain regions at the same time. The system works alongside tracing techniques that map the connections between distant neurons.

To demonstrate the new tools, Chen, Voigt et al. measured the activity of neurons in two areas of the mouse brain that monitor the whiskers. Mice brush their whiskers against an object to obtain information on its size, shape, texture and location. Two brain regions, called the primary and secondary areas of the whisker cortex, process this information and exchange messages back and forth. However, it was unclear what information these messages contain.

Chen, Voigt et al. therefore trained mice to discriminate between coarse and fine sandpapers using their whiskers, and analysed the activity of the neurons that directly connect the two areas of the whisker cortex. The results revealed that although movement and sensory stimulation activated both the primary and secondary areas of the whisker cortex, the direct connections between these regions mainly exchange sensory information.

This approach makes it possible to observe brain networks in an unprecedented level of detail. In the future, this technology will be extended to provide a more comprehensive view of how neurons communicate across brain areas. This will increase our understanding of how multiple areas of the brain all work together to produce the activity patterns that give rise to behavior.

to attribute such activity to anatomically-defined neuronal subsets. Consequently, it has not been possible to definitively determine whether the underlying measured inter-areal dynamics could reflect: i) direct cortico-cortical interactions; ii) indirect cortico-thalamocortical pathways; iii) or synaptic drive from common input areas. To dissect these possibilities new technologies are needed to monitor inter-areal dynamics with cellular resolution while at the same time identifying subsets of neurons that project across areas. Two-photon microscopy is well suited to monitor action potential firing across neuronal populations, mainly using calcium imaging, as well as to optically identify molecularly or anatomically-defined cell types (*Chen et al., 2013a*). So far, standard two-photon microscopes have been limited to imaging long-range activity within one cortical area (*Chen et al., 2013b*; *Glickfeld et al., 2013*; *Jarosiewicz et al., 2012*; *Petreanu et al., 2012*; *Sato and Svoboda, 2010*). New systems have recently been developed that enable simultaneous imaging of neuronal populations across cortical areas across increasingly larger fields of view (*Lecoq et al., 2014*; *Stirman et al., 2014*; *Tsai et al., 2015*).

Here, we present a novel 'multi-area' two-photon microscope for simultaneous measurements across primary and higher sensory areas of mouse neocortex. We have combined this system with anatomical labeling strategies to identify feedforward and feedback projection neurons between reciprocally connected cortical areas to image their functional interactions. In order to investigate the role of direct cortico-cortical interactions among other potential sources of correlated activity, we have applied this approach in the whisker primary (S1) and secondary (S2) somatosensory cortices, two areas that are anatomically coupled through reciprocal connections, cortico-thalamocortical pathways, and other common inputs (*Deschenes et al., 1998*; *Suter and Shepherd, 2015*; *Theyel et al., 2010*). Expanding our recent work on the activity of divergent projection pathways originating in S1 during a texture discrimination task (*Chen et al., 2013b*; *2015*), we sought here to examine how population activity in S1 and S2 evolves over time during such tactile whisker-based behavior. Whisking behavior spans a range of time scales, from individual whisk cycles of about 100-

ms duration, to bouts of whisking over a second, and to prolonged whisking, for example during locomotion (*Kleinfeld and Deschenes, 2011*). Our multi-area imaging approach enabled us to analyze the slower aspects of whisking envelope changes and whisker-touch contacts whereas analysis of neuronal dynamics on the rapid time scale of tens of milliseconds was precluded by our limited temporal resolution. Our main goal was, however, to take advantage of the ability to simultaneously image in S1 and S2 and to investigate how the subsets of reciprocally projecting neurons contribute to the coordination of activity across these areas and to the coding of sensory and behavior information.

## Results

### Multi-area two-photon microscope

We built a two-photon microscope capable of simultaneous scanning of two sub-areas within a relatively large field of view (FOV), enabling one to freely and independently position the sub-areas in order to select appropriate imaging spots. To achieve this goal we coupled two laser beams through a galvanometric scanner system into a low-magnification, high-NA objective (*Figure 1A–D* and Materials and methods). Specifically, we chose a 16x water-immersion Nikon objective (NA 0.8) as core element, which supports imaging in a FOV of 1.8-mm maximum side length with cellular resolution (*Figure 1E*, *Figure 1—figure supplement 1* and *Video 1*). We split laser light from a Ti:sapphire laser (80 MHz pulse repetition rate) into two excitation beams using a 50:50 beam splitter and delayed the laser pulse train of one beam by 6.25 ns, half of the inter-pulse interval, to interlace the two pulse trains so that the two sub-areas receive alternating laser excitation pulses. For disambiguating the fluorescence signal generated by the two laser foci, we adopted a rapid de-multiplexing approach (*Cheng et al., 2011*). For typical 2–4 ns fluorescence lifetimes of fluorescent proteins (*Akerboom et al., 2013*), the 6.25-ns time windows are sufficiently long to capture mainly fluorescence photons generated by the last excitation pulse. Some crosstalk between areas may remain but can be corrected for post hoc using spatial linear unmixing (Materials and methods and *Figure 1—figure supplement 2*) (*Cheng et al., 2011*).

Each beam enters a movable coupling unit, named 'focal plane unit' (FPU), which enables independent positioning and focusing of its respective imaging area below the objective (*Figure 1A–C*). Independent positioning is achieved by coupling the FPU output beams to the scanner unit via small fold mirrors that sit at the end of cantilever arms. Lateral x/y-movement of each FPU introduces an offset of the respective beam from the optical axis of the first scan lens, which converts this offset into a pivoting angle of the beam around the scan mirrors. In the remaining optical path, this pivoting angle is translated into lateral movement of the corresponding imaging sub-area below the objective. Independent focusing is achieved with electrically tunable lenses (ETLs) in the FPUs (*Grewe et al., 2011*). Each ETL is combined with an offset lens to allow tuning the beam from divergent to convergent. These divergence changes translate into axial shifts of the intermediate foci at the FPU output and in between scan and tube lens, corresponding to down- and upward shifts of the excitation focus along the optical z-axis below the objective. In combination with a 6-mm pair of scan mirrors, ETL focusing provides a z-range of up to 600 μm.

### Imaging anatomically-identified projection neurons across S1 and S2

Mice can actively sense the environment by moving their whiskers to gather information regarding the location, shape, size, and texture of an object (*Diamond et al., 2008*). Processing of tactile information at the cortical level is thought to occur through interactions between S1, S2, as well as primary motor cortex (M1) (*Aronoff et al., 2010*; *Bosman et al., 2011*). In order to investigate direct interactions between S1 and S2, we sought to apply the multi-area two-photon microscope to simultaneously monitor activity in feedforward neurons in S1 projecting to S2 ($S1_{S2}$) and feedback neurons in S2 projecting to S1 ($S2_{S1}$) in wild-type adult mice during tactile whisker behavior. To distinctly label these projection neurons in a mutually exclusive manner across the reciprocally connected areas we employed a viral strategy making use of orthogonal recombinase systems. To label $S2_{S1}$ neurons, we delivered a retrogradely-infecting AAV6 expressing Cre recombinase (AAV6-*pgk-Cre*) into S1 along with S2-injection of an AAV1 expressing Cre-dependent nuclear tdTomato (AAV1-*EF1α-dio-NLStdTomato*; *Figure 2A*). $S1_{S2}$ neurons were labeled by delivering an AAV6 expressing

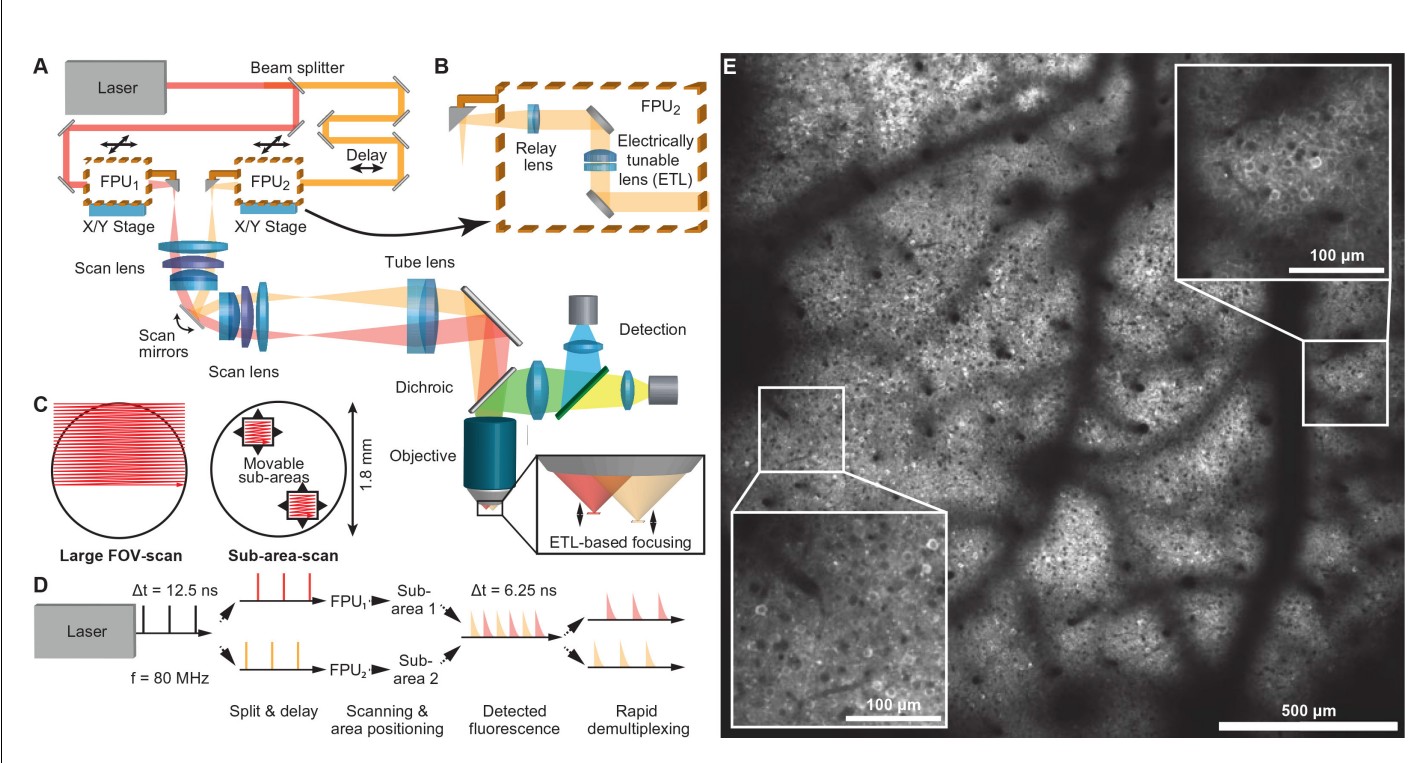

**Figure 1.** Multi-area two-photon microscope for flexible simultaneous imaging of sub-areas within a large field-of-view. (**A**) Schematic of multi-area two-photon microscope. Light from a Ti:sapphire laser is split into two beams and one beam sent to a delay line. Each beam then enters a focal plane unit (FPU), which allows axial focusing with an electrically tunable lens (ETL). Both beams are scanned in parallel by a pair of galvo mirrors. (**B**) Schematic of FPU. (**C**) Imaging modes include scanning of a single large FOV (with one beam switched off) and parallel scanning of two sub-areas. (**D**) Principle of spatiotemporal multiplexing: The detected fluorescence photons can be attributed to the correct area of origin by rapid demultiplexing synchronized to the laser pulse train. (**E**) Example two-photon image (1.7 mm FOV) at 160–180 μm depth in a YCX2.60-expressing transgenic mouse in L2/3.

The following figure supplements are available for figure 1:

**Figure supplement 1.** Variation of the point-spread function over field-of-view position and ETL tuning range.

**Figure supplement 2.** Crosstalk between both sub-areas observed in vivo.

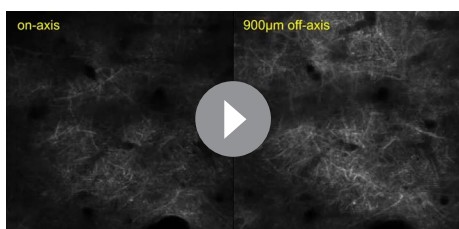

**Video 1.** In vivo z-stack of YC-Nano140 expressing neurons. Single area images from the multi-area two-photon microscope of L2/3 neurons were taken from 70-210 μm below the pial surface at 1 μm z-step resolution. Sub-area excitation beam was delivered through the ETL, positioned either on-axis (left) or 900 μm off-axis (right), and focusing was achieved through translation of the objective by the z-stage.

Flpe (AAV6-*syn-Flpe*) into S2 along with S1-injection of an AAV1 expressing Flpe-dependent nuclear LSSmKate2 (AAV1-*EF1α-fio-H2BLSSmKate2*). In addition to these viruses, we broadly expressed the genetically encoded calcium indicator YC-Nano140 in S1 and S2 using AAV1-*EF1α-YCNano140* (*Chen et al., 2013b*; *Horikawa et al., 2010*). For targeting viral injections as well as for selecting regions for later two-photon imaging, we employed optical intrinsic signal imaging to identify areas within S1 and S2 corresponding to the same principal whisker (*Figure 2B* and Materials and methods).

Following cranial window implantation, LSSmKate2-positive S1$_{S2}$ neurons and tdTomato-positive S2$_{S1}$ neurons in L2/3 were identified in vivo (*Figure 2C*). YC-Nano140 expressing neurons that did not express LSSmKate2 or

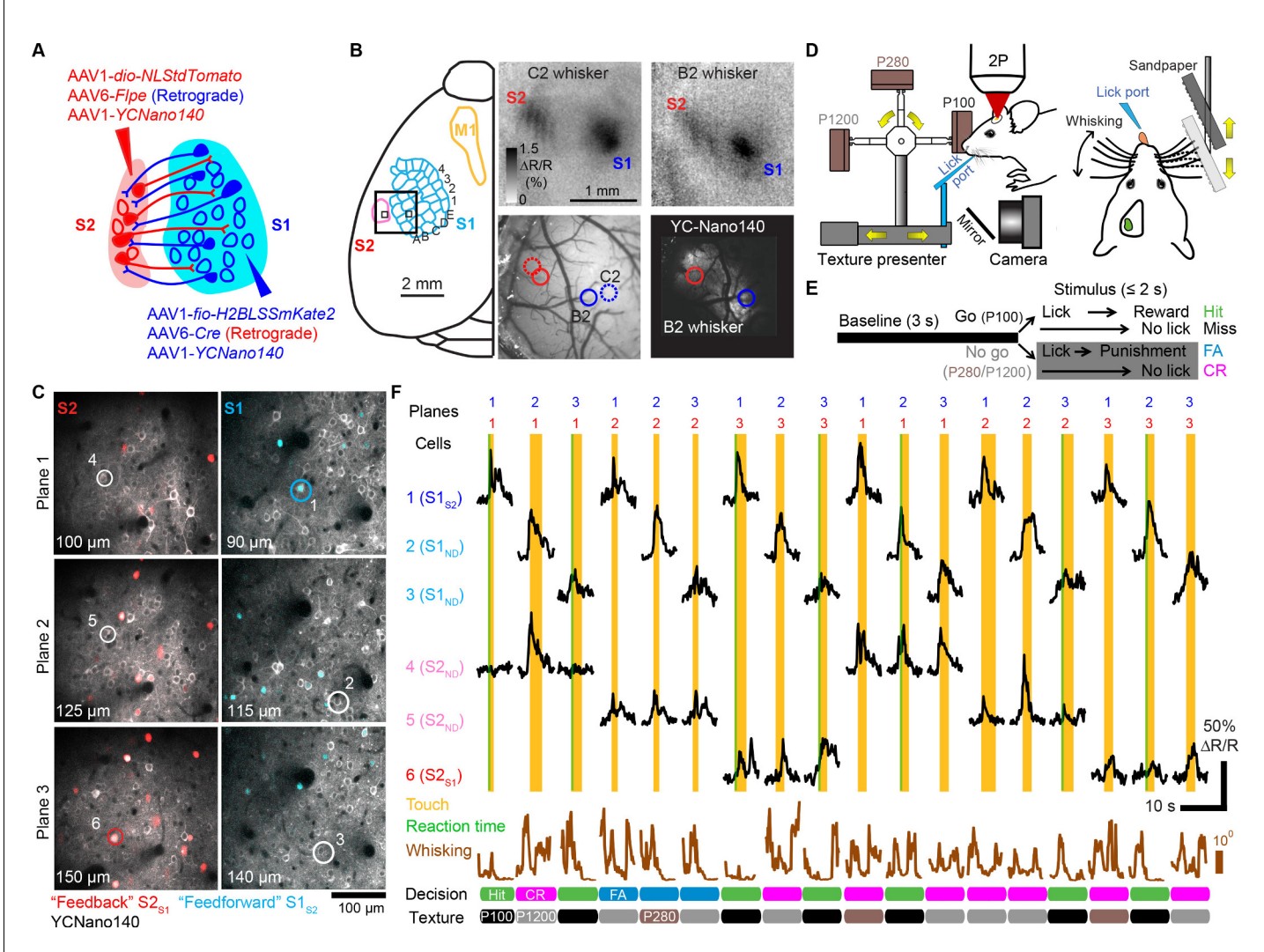

**Figure 2.** Simultaneous calcium imaging of identified feedforward and feedback neurons in S1 and S2 of mouse neocortex during behavior. (**A**) Viral injection scheme for simultaneous labeling of feedforward and feedback neurons and YC-Nano140 expression. (**B**) Functional mapping of S1 and S2 through optical intrinsic signal imaging. Intrinsic signals evoked by stimulation of the C2 whisker (top left) and the B2 whisker (top right). In addition to localized intrinsic signals in S1 barrel columns, additional activation spots are visible in S2. Identified barrel columns (circles) are overlaid over blood vessel (bottom left) and YC-Nano140 expression (bottom right) images. (**C**) In vivo 2-photon images of LSSmKate2-positive $S1_{S2}$ neurons (blue), tdTomato-positive $S2_{S1}$ neurons (red) with non-co-labeled YC-Nano140-expressing neurons (grey) in S1 ($S1_{ND}$) and S2 ($S2_{ND}$). (**D**) Behavior setup for texture discrimination task. (**E**) Trial structure for go/no-go texture discrimination task. (**F**) Example calcium transients for individual neurons in [C] measured episodically during texture discrimination task along with periods of whisker-to-texture touch (orange area), whisking amplitude (brown trace), and reaction time on Hit trials (green area). For each trial the selected plane in each sub-area is indicated on top, illustrating the combinatorial plane hopping.

The following source data and figure supplement are available for figure 2:

**Source data 1.** Optimized low tensor rank across animals.

**Figure supplement 1.** Denoising with tensor decomposition.

tdTomato were classified as $S1_{ND}$ and $S2_{ND}$ neurons, respectively (target area 'not determined'), possibly comprising unlabeled $S1_{S2}$ and $S2_{S1}$ neurons as well as projection neurons targeting different brain regions. Animals were habituated to head-fixation and trained to perform a whisker-based go/no-go texture discrimination task (*Figure 2D,E*) (*Chen et al., 2013b*; *2015*). On 'go' trials,

animals were rewarded with a water droplet if they correctly licked ('Hit') when presented with a target texture (a panel of coarse sandpaper, P100). On 'no-go' trials, mice were supposed to withhold licking ('correct rejection' or 'CR') when presented with one of two non-rewarded, 'non-target' textures of smoother grades (P280, P1200). Misses on go trials were not rewarded and false alarms ('FA') on no-go trials were punished with an air puff and a time-out period. Whisker movements were monitored with high-speed videography (500 Hz) and licking behavior was measured with a piezo film attached to the water spout. Whisking and licking recordings were downsampled to match the frame rate of imaging (7 Hz), allowing analysis of how neuronal activity relates to slow amplitude changes of whisking envelope and to the occurrence of whisker-texture touches (Materials and methods).

Since simultaneous imaging in two cortical regions presents unique opportunities to examine the coordination of activity across areas, we sought to increase the number of pairwise imaged S1 and S2 neuronal populations. To this end we used the ETLs to implement a 'combinatorial plane hopping' mode, in which two sub-areas are scanned simultaneously but each imaging plane is independently refocused in a combinatorial manner during the inter-trial interval (*Figure 2C,F* and *Video 2*). Using this approach, we imaged in 7 mice ~150 neurons per sub-area (distributed over three imaging planes at different cortical depths) across ~1800 trials over 5–6 experimental sessions. Combinatorial hopping between three imaging planes in each area resulted in simultaneous imaging of 9 combinations of planes per animal, for which ~200 trials were acquired per pair of planes, still sufficient for our analysis. In total, 228 $S1_{S2}$, 817 $S1_{ND}$, 193 $S2_{S1}$, and 750 $S2_{ND}$ neurons were imaged in 63 pairs of focal planes across S1 and S2. For comparison with non-task conditions, we additionally imaged the same neuronal populations as measured during texture discrimination behavior for another ~1800 trials over 5–6 sessions, during which mice were passively presented with the same textures. In order to improve statistical analysis of single-trial responses for trial conditions with low trial numbers, calcium traces were denoised using tensor decomposition (*Figure 2—figure supplement 1* and *Figure 2—source data 1*, Materials and methods).

## Behavior-related responses of S1 and S2 neurons

While sensory- and behavior-related responses of $S1_{S2}$, $S1_{M1}$, and $S1_{ND}$ neurons have been characterized during texture discrimination (*Chen et al., 2013b*; *2015*), responses of $S2_{S1}$ and $S2_{ND}$ neurons have not. We first assessed for each cell class how calcium signals relate to behavioral aspects using a general linear model (GLM) against vectors for whisker-touch onset, whisking envelope amplitude, and licking onset (*Figure 3A,B* and *Figure 3—figure supplement 1*) (*Miri et al., 2011*; *Pinto and Dan, 2015*). $S1_{ND}$ and $S2_{S1}$ neurons showed better overall GLM fits to these behavioral parameters compared to their neuronal counterparts in their respective areas (*Figure 3C* and *Figure 3—figure supplement 1*; $S1_{ND}$ vs. $S1_{S2}$, p<0.002; $S2_{ND}$ vs. $S2_{S1}$, p<0.005, KS-test). Further analysis of fits to specific regressors revealed that $S2_{S1}$ and $S1_{ND}$ neurons showed higher GLM coefficients for whisker-touch onset than their within-area counterparts (*Figure 3D*; $S1_{ND}$ vs. $S1_{S2}$, p<0.05; $S2_{ND}$ vs. $S2_{S1}$, p<0.005, one-way ANOVA with repeated measures). While no specific differences were observed for cell classes in S1, $S2_{S1}$ neurons showed higher GLM coefficients than $S2_{ND}$ neurons for whisking and licking onset (p<0.001, one-way ANOVA with repeated measures). These results suggest that $S2_{S1}$ neurons exhibit higher whisking- and licking-related activity compared to other neurons in S2.

We next analyzed single-neuron responses to different sensory conditions or different behavior conditions by performing single-cell receiver operating characteristic (ROC) analysis against different trial conditions (*Green and Swets, 1966*). Single-cell ROC analysis of Hit vs. CR trials revealed that a larger fraction of $S2_{S1}$ neurons compared to other neuronal classes (72%) was able to discriminate these two conditions above chance (*Figure 3E*; p<0.002, $\chi^2$ test).

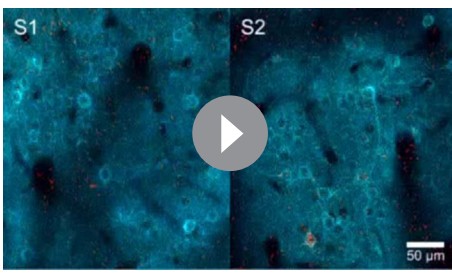

**Video 2.** Simultaneous calcium imaging across S1 and S2. Single trial video of calcium responses during texture discrimination acquired at 7 Hz with the multi-area two-photon microscope (1x playback speed). YFP (green) and CFP (blue) fluorescence from YC-Nano140 are shown and overlaid with calculated $\Delta R/R$ (red).

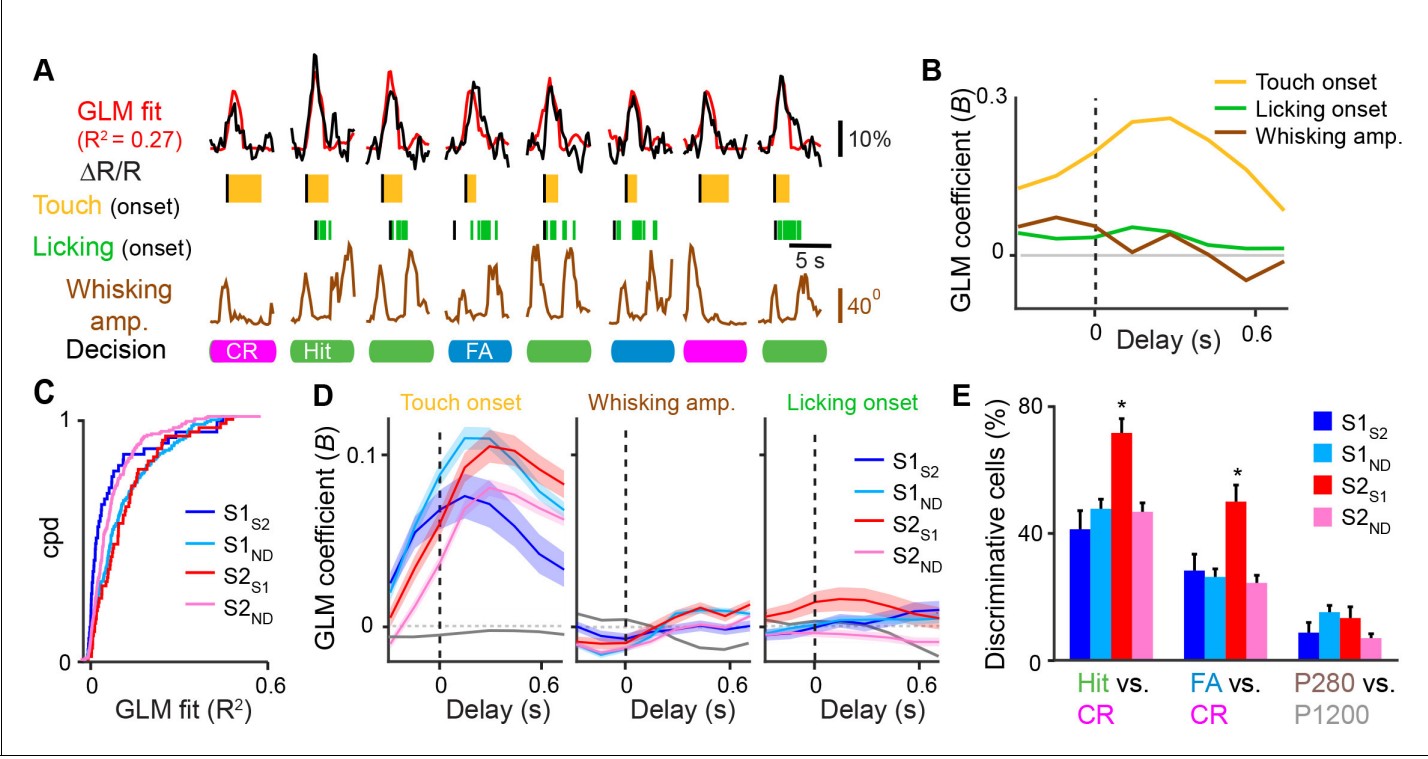

**Figure 3.** Feedback neurons in S2 exhibit behavior-related responses. (**A**) General linear model (GLM) of behavior-related responses. Example of GLM fit for one neuron of calcium responses against touch, licking, and whisking as behavior events. Single-trial calcium responses are plotted along with model fit as well as touch periods with onset indicated, individual licks with onset indicated, whisking envelope amplitude, and decision. (**B**) GLM coefficients (*B*) for example neuron shown in [A] for regressors for touch onset, whisking envelope amplitude, and licking onset across different delays. Delays are aligned to the onset of each behavioral event. (**C**) Cumulative probability distribution (cpd) of overall GLM fit across cell types. (**D**) GLM coefficients for different cell types for regressors for touch onset (left), whisking envelope amplitude (middle), and licking onset (right) across different delays. Grey line indicates average GLM coefficient for neurons with non-significant coefficients at that time point. (**E**) Fraction of active neurons able to discriminate Hit vs. CR, FA vs. CR, and P280 vs. P1200 trials above chance determined by single-cell ROC analysis. (shaded area: s.e.m. error bars: s.d. from bootstrap test; *n* = 44 $S1_{S2}$, 161 $S1_{ND}$, 59 $S2_{S1}$, 198 $S2_{ND}$ neurons).

The following figure supplement is available for figure 3:

**Figure supplement 1.** General linear model of whisking- and licking-related calcium responses.

Differences between Hit vs. CR trials could reflect encoding of sensory information, decision, or decision-related actions such as licking. To disambiguate these possibilities, we also performed ROC analysis of FA vs. CR trials, which were previously shown to consist of similar whisking and sensory conditions (*Chen et al., 2015*). Again, a larger fraction of $S2_{S1}$ neurons (50%) was able to discriminate these two conditions above chance (p<0.05, $\chi^2$ test), suggesting that this greater discrimination power of $S2_{S1}$ neurons represents decision- or action-related information. As an additional control, we assessed sensory-related responses by ROC analysis of P280 vs. P1200 textures on CR trials and found that $S2_{S1}$ neurons were not more likely to discriminate these trial types compared to other cell types (*Figure 3E*). Overall, we find that $S2_{S1}$ were more likely to encode for non-sensory aspects of task-related behavior compared to other neurons in S2 and S1.

## Motor-related coordination of S1 and S2

In order for task-related information exchange to occur between areas, activity across areas must be 'coordinated' during relevant behavioral conditions and such coordination should be specific to neurons that anatomically project between areas. To investigate how activity is coordinated across S1 and S2, we first sought a measure of population activity for each area that would capture the diverse response properties of individual neurons and allow us to determine if their dynamics evolve similarly

across time. To this end, we characterized population activity in S1 and S2, respectively, by using linear discriminant analysis (LDA) (*Fisher, 1936*; *Safaai et al., 2013*). For *n* neurons in an imaging area, LDA finds for each time point an axis in *n*-dimensional space so that the distributions of population responses for two chosen trial conditions – projected onto this axis – are best separated (Materials and methods). Similar to the ROC analysis, we selected not only Hit vs. CR but also various other pairs of trial conditions that would allow us to disambiguate sensory- and behavior-related dynamics (*Table 1*). The dimensionality reduction resulting from this approach effectively extracts time-dependent 'linear discriminant' variables LD($t$) as one-dimensional representations of neuronal population activity with respect to the chosen trial conditions. For illustration purposes, we exemplify this LDA procedure for measurements from only two neurons in *Figure 4A,B*, whereas typically populations of active neurons within an imaging area were used for analysis.

In the initial analysis of population responses, we did not distinguish between neuronal cell types in each area and thus included both $S1_{S2}$ and $S1_{ND}$ neurons for S1 and $S2_{S1}$ and $S2_{ND}$ neurons for S2. We performed LDA at each time point for 1-s periods prior to and following either whisker-touch onset or licking onset, generating mean LD time courses for S1 or S2 by averaging LDA results from all imaging areas in these respective regions. For LDA performed on Hit vs. CR trials, we observed that mean population responses for both S1 and S2 diverged following whisker-touch onset (*Figure 4C*). ROC analysis using the LD variable as measure of population activity in S1 and S2, respectively, revealed that discrimination power for both areas increased immediately following whisker-touch onset and through the first second of touch (*Figure 4D*).

S1 and S2 receive common input from several areas including M1, which controls licking and whisking (*Brecht et al., 2004*; *Huber et al., 2012*; *Suter and Shepherd, 2015*), the posteromedial thalamic nucleus (POm), which relays re-afferent whisking (*Deschenes et al., 1998*; *Moore et al., 2015*; *Yu et al., 2006*), and the ventral lateral region of the ventral posterior medial thalamic nucleus (VPMvl), which relays whisker touch (*Pierret et al., 2000*). Correlations of activity between S1 and S2 could thus reflect these aspects of behavior. To measure how S1 and S2 activities are coordinated across time, we calculated the trial-by-trial correlation of $LD_{S1}$ and $LD_{S2}$, the LD time courses obtained for active neurons in simultaneously imaged populations in S1 and S2, respectively (*Figure 5A*). We termed this cross-areal correlation $LDCC_{S1:S2}$, which during task performance increased immediately following whisker-touch onset for both Hit and CR trials. 500 ms after touch onset, however, $LDCC_{S1:S2}$ remained elevated for Hit trials relative to CR trials (*Figure 5B*, $p<0.05$, one-way ANOVA with repeated measures). The time point of this divergence corresponded to the average delay of licking onset from whisker-touch onset (mean reaction time: $524 \pm 5$ ms for Hit trials; *Figure 5C*) (*Chen et al., 2013b*; *2015*). To examine whether $LDCC_{S1:S2}$ changes indeed relate to the reaction time, $LDCC_{S1:S2}$ on Hit trials were re-aligned to licking onset (*Figure 5D*). $LDCC_{S1:S2}$ increased and peaked at licking onset and remained elevated thereafter, further suggesting that coordination of activity across S1 and S2 could be associated with such behavior.

To dissociate whether and how cross-areal coordination related to sensory versus motor parameters, we first controlled for sensory input by measuring $LDCC_{S1:S2}$ for population responses projected along the FA vs. CR axis, where the same non-target textures were presented (*Figure 5E* and *Figure 5—figure supplement 1A,B*). We found $LDCC_{S1:S2}$ was higher for FA compared to CR trials

**Table 1.** Axes used for linear discriminant analysis. Summary of trial conditions compared and used for linear discriminant analysis. For each axes, noted are potential differences in texture, licking, and whisking parameters between trial conditions as well as the utility in comparing such trial conditions for isolating sensory or behavior responses.

| Axes for LDA | Texture | Licking | Whisking | Utility in analysis |
|---|---|---|---|---|
| Hit vs. CR | Different | Different (Hit) | Same | Cannot isolate sensory, decision, or action-related responses |
| FA vs. CR | Same | Different (FA) | Same | Isolate decision and action-related responses |
| Pre- vs. post-touch licking (FA trials) | Same | Different | Same | Isolate licking-related responses |
| High vs. Low Whisking (CR trials) | Same | None | Different | Isolate whisking-related responses |
| P280 vs. P1200 (CR trials) | Different | None | Same | Isolate sensory-related responses |
| Target vs. Non-target (Non-task) | Different | None | None | Isolate sensory-related responses |

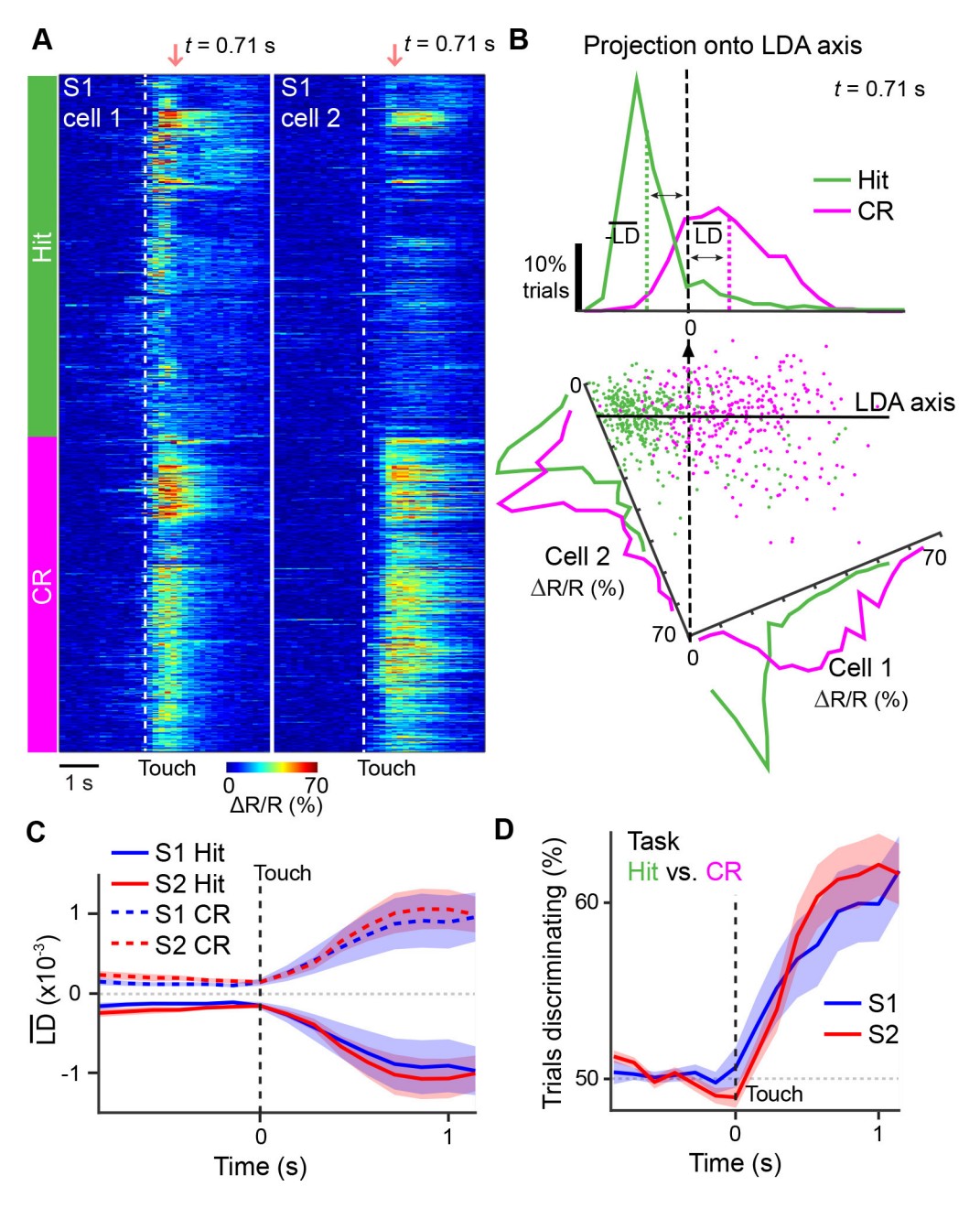

**Figure 4.** Illustration of extracting population response time courses by linear discriminant analysis. (**A**) While LDA is performed on multiple simultaneously imaged neurons, for demonstration purposes, here calcium transients of two simultaneously imaged neurons within an imaging plane are plotted and sorted according to Hit and CR trials. Dotted line indicates whisker-touch onset. (**B**) Example linear discriminant analysis performed on the two neurons in [**A**]. Bottom panel shows scatter plot of trial-by-trial responses for each neuron at the indicated time point (red region in [**A**]) rotated along the LD axis for Hit vs. CR trials. Top panel shows distribution of trials for population activity projected along the LD axis along with mean LD response. (**C**) Average S1 or S2 population responses after LDA in Hit and CR trials across the first second prior to and following whisker-touch onset. (**D**), ROC analysis of S1 or S2 population responses shown in [**C**] for Hit vs. CR trials under task conditions demonstrating the performance of the LDA. Dotted line indicates touch onset. (shaded area: s.e.m.; *n* = 21 S1 planes, 21 S2 planes).

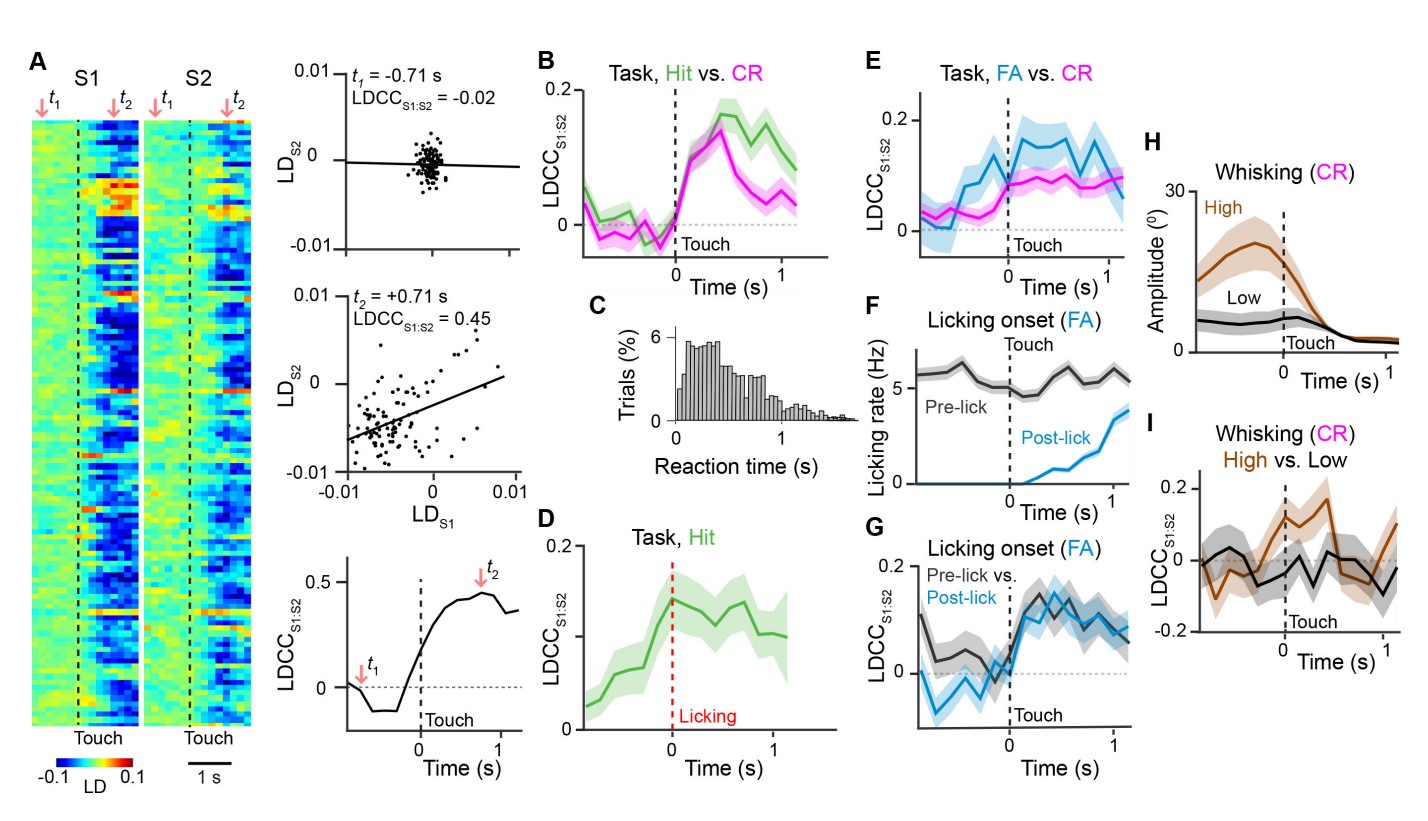

**Figure 5.** Motor behavior is associated with coordinated population activity across S1 and S2. (A) Analysis of coordinated activity across S1 and S2. Left panel shows example of single-trial population responses for Hit trials projected along Hit vs. CR axis for simultaneously imaged S1 (LD$_{S1}$) and S2 (LD$_{S2}$) of S2 sub-areas. Upper right panels shows trial-by-trial correlations (LDCC$_{S1:S2}$) between LD$_{S1}$ and LD$_{S2}$ at indicated time points. Bottom right panel shows calculated LDCC$_{S1:S2}$ across the trial period. (B) LDCC$_{S1:S2}$ for Hit vs. CR trials. (C) Normalized histogram of reaction times across Hit trials. (D) LDCC$_{S1:S2}$ for Hit trials along Hit vs. CR axis aligned to licking onset. (E) LDCC$_{S1:S2}$ for FA vs. CR trials. (F) Licking rate for FA trials in which licking onset precedes (pre-wo) and follows (post-wo) whisker-touch onset. (G) LDCC$_{S1:S2}$ for pre-wo vs. post-wo FA trials. (H) High vs. low whisking amplitude CR trials. (I) LDCC$_{S1:S2}$ for high vs. low whisking amplitude CR trials. All time course data are aligned to whisker-touch onset (black dotted line, x-axis) except for [D] which is aligned to licking onset (red dotted line). shaded area: s.e.m.; (C,D,E,G,I) n = 63 pairs of S1 and S2 planes in 7 animals; (C) n = 7120 trials (F) n = 1120 trials (H) n = 7 animals, 6960 trials.

The following figure supplements are available for figure 5:

**Figure supplement 1.** Linear discriminant analysis across different sensory or behavior axes.

**Figure supplement 2.** Coordinated activity across S1 and S2 is not stimulus-specific.

both prior to and following whisker-touch onset, (p<0.05, one-way ANOVA with repeated measures). We asked if this increased LDCC$_{S1:S2}$ on FA trials could partially be explained by licking behavior. We therefore subdivided FA trials into trials in which licking preceded whisker-touch onset – likely reflecting impulsive licking behavior – and those trials in which licking onset occurred after whisker-touch onset (67.1% and 32.9% of FA trials, respectively; *Figure 5F*). LDCC$_{S1:S2}$ showed an increased level prior to whisker-touch onset specifically for the subset of trials with early licking (*Figure 5G,*p<0.05, one-way ANOVA with repeated measures). This suggests that population activity in S1 and S2 can be coordinated during licking behavior both in the presence and absence of sensory stimulation.

We next asked whether LDCC$_{S1:S2}$ is also related to other motor behaviors such as whisking. During texture discrimination, animals adopted a high-amplitude, rhythmic whisking strategy prior to whisker-touch onset in anticipation of the delivered texture, which drives texture-specific kinematics and is absent in non-task sessions (*Chen et al., 2013b*; *2015*). During task conditions, we measured

$LDCC_{S1:S2}$ from population responses for CR trials (i.e. same texture, no licking) projected along the high- vs. low-amplitude whisking axis (*Figure 5H* and *Figure 5—figure supplement 1C–D*). Similar to the results for licking behavior, high-amplitude whisking trials were associated with higher $LDCC_{S1:S2}$ prior to and after whisker-touch onset when compared to low-amplitude whisking trials (*Figure 5I,*p<0.02, one-way ANOVA with repeated measures), demonstrating another motor-related component of S1-S2 coordination. By using LDA for other pairs of trial conditions, we found that stimulation with distinct textures did not result in elevated $LDCC_{S1:S2}$, suggesting that S1-S2 coordination is not stimulus-specific (*Figure 5—figure supplement 1G–H* and *Figure 5—figure supplement 2*). Taken together, this demonstrates that the coordination of population activity across S1 and S2 can be associated with licking and whisking behavior that is independent of sensory stimulus.

## Projection neurons contribute to coordinated activity

Correlated changes in population dynamics across cortical areas can either reflect direct cortico-cortical interactions, indirect interactions through cortico-thalamocortical pathways, or co-activation from another common input source (*Salinas and Sejnowski, 2001*). In order for direct cortico-cortical interactions to be present, such correlations should exist in neurons that project between those areas. To understand how $S1_{S2}$ and $S2_{S1}$ neurons might contribute to the coordination of population activity in S1 and S2, we repeated the LDA for S1 or S2 but shuffled the trial-by-trial responses of $S1_{S2}$ and $S2_{S1}$ neurons when projecting the population response onto the LD axis (*Figure 6—figure supplement 1A*). In order to ensure that changes in the population response were specific to these neurons and not merely a result of altering any given subpopulation of neurons, we also computed population responses, in which trials from an equal number of $S1_{ND}$ and $S2_{ND}$ neurons were shuffled (see details in Materials and methods). We observed no significance difference in the trajectory or discrimination power of S1 and S2 population responses when shuffling any of these cell types (*Figure 6—figure supplement 2*), suggesting that the average population response within each area was not altered with this analysis.

To determine the specific contribution of $S1_{S2}$ and $S2_{S1}$ to inter-areal coordination, we measured the change in correlation between the S1 and S2 population responses ($\triangle LDCC_{S1:S2}$; relative to unshuffled controls) that resulted from shuffling trials of these projection neurons and compared it to the result of shuffling trials of $S1_{ND}$ and $S2_{ND}$ neurons (*Figure 6A* and *Figure 6—figure supplement 1B*). If $S2_{S1}$ and $S1_{S2}$ neurons especially contribute to $LDCC_{S1:S2}$, their trial-shuffling should lead to a larger reduction (more negative $\triangle LDCC_{S1:S2}$) compared to trial-shuffling $S1_{ND}$ and $S2_{ND}$ neurons. Analysis of coordinated activity projected along the Hit vs. CR axis showed no significant difference in $\triangle LDCC_{S1:S2}$ between $S2_{S1}$ and $S1_{S2}$ neurons and $S2_{ND}$ and $S1_{ND}$ neurons when aligned to whisker-touch onset (*Figure 6—figure supplement 3*). However, analysis of Hit trials after aligning to licking onset revealed a negative dip in $\triangle LDCC_{S1:S2}$ when shuffling projection neurons, indicating that $S1_{S2}$ and $S2_{S1}$ neurons especially contributed to $LDCC_{S1:S2}$ upon licking onset (*Figure 6B*, p<0.0001, one-way ANOVA with repeated measures).

We further assessed the contribution of $S2_{S1}$ and $S1_{S2}$ neurons to inter-areal coordination along motor conditions by analyzing whisking- and licking-related $LDCC_{S1:S2}$. We first measured $\triangle LDCC_{S1:S2}$ from population responses projected onto the high- vs. low-whisking amplitude axis (for CR trials) and found that $S1_{S2}$ and $S2_{S1}$ neurons significantly contributed to $LDCC_{S1:S2}$ in high-amplitude whisking trials following but not preceding whisker-touch onset (*Figure 6C*, p<0.02, one-way ANOVA with repeated measures). Similarly, analysis of FA vs. CR trials showed that $S2_{S1}$ and $S1_{S2}$ neurons did not specially contribute to $LDCC_{S1:S2}$ for FA trials in which licking onset preceded whisker-touch onset (*Figure 6D*), but they did so for trials in which licking onset followed whisker-touch onset (*Figure 6E*, p<0.02, one-way ANOVA with repeated measures). These findings indicate that while licking and whisking behavior is associated with correlations in population responses across S1 and S2, any special contribution of $S1_{S2}$ and $S2_{S1}$ neurons to this coordination depends on the presence of sensory stimulus, thus occurring after the whisker-touch onset. Hence, the specific contribution of projection neurons mutually connecting S1 and S2 could reflect sensory- rather than motor-related activity. In line with this notion, further analysis showed that $\triangle LDCC_{S1:S2}$ decreased when trial-shuffling $S1_{S2}$ and $S2_{S1}$ neurons compared to $S1_{ND}$ and $S2_{ND}$ neurons following whisker-touch onset on CR trials, when licking behavior is absent (*Figure 6F*, p<0.02, one-way ANOVA with repeated measures). These results demonstrate that direct cortico-cortical interactions through $S1_{S2}$ and $S2_{S1}$ neurons reflect exchange of sensory or decision information rather than motor information.

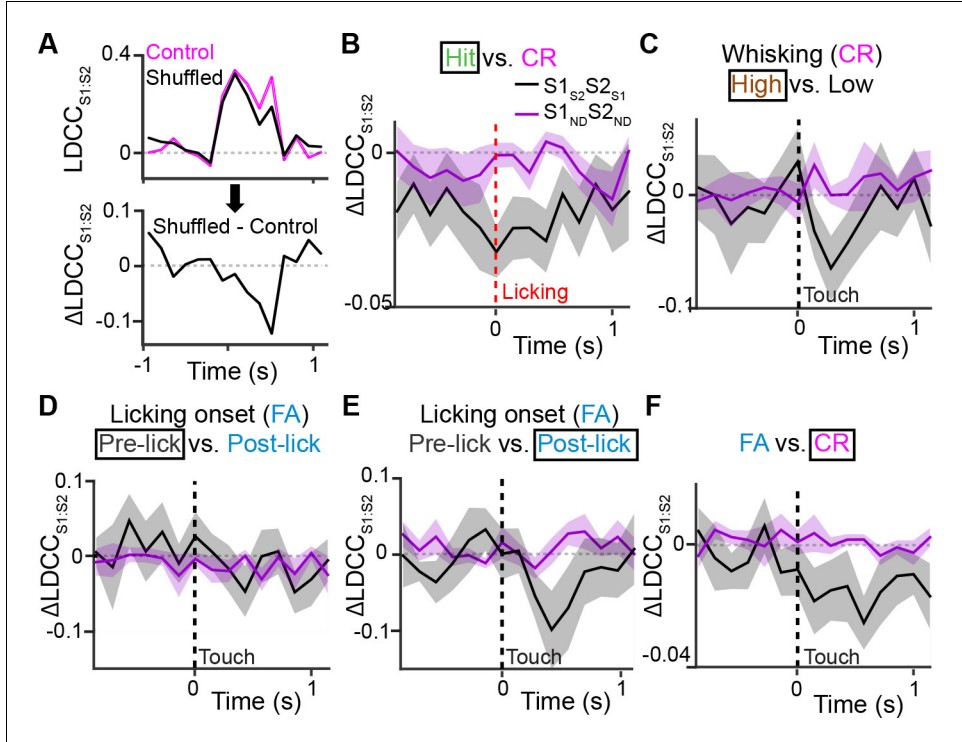

**Figure 6.** Projection neurons contribute to coordinated S1 and S2 activity. (A) The contribution of specific cell types to coordinated activity across S1 and S2 is measured by trial-shuffling responses for those cell types prior to calculating the LDCC$_{S1:S2}$. The resulting LDCC$_{S1:S2}$ from the shuffled condition is then subtracted by the LDCC$_{S1:S2}$ from the control condition to obtain ΔLDCC$_{S1:S2}$ (see also ***Figure 6—figure supplement 1***). (B), ΔLDCC$_{S1:S2}$ for Hit trials along the Hit vs. CR axis after aligning to licking onset. (C) ΔLDCC$_{S1:S2}$ for high-amplitude whisking CR trials along the high vs. low whisking amplitude CR trial axis. (D) ΔLDCC$_{S1:S2}$ for FA trials, in which licking onset precedes whisker-touch onset along the FA vs. CR axis. (E) ΔLDCC$_{S1:S2}$ for FA trials, in which licking onset follows whisker-touch onset along the FA vs. CR axis. (F) ΔLDCC$_{S1:S2}$ for CR trials along the FA vs. CR axis. All time course data are aligned to whisker-touch onset (dotted line, x-axis) except for [B] which is aligned to licking onset (red dotted line). (shaded area: s.e.m.; *n* = 21 S1 planes, 21 S2 planes, 63 pairs of S1 and S2 planes in 7 animals).

The following figure supplements are available for figure 6:

**Figure supplement 1.** Measuring the contribution of specific cell types to coordinated population activity.

**Figure supplement 2.** Projection of shuffled trials does not alter average population response.

**Figure supplement 3.** Contribution of S1S2 and S2S1 neurons to Hit and CR trials relative to whisker-touch onset.

## Discussion

In summary, we have demonstrated simultaneous measurement of calcium signals in identified feed-forward and feedback neurons across S1 and S2 in the awake behaving mouse using a multi-area two-photon microscope in combination with viral-mediated labeling of long-range projection neurons. We have used this approach to investigate the contribution of cortico-cortical projection neurons to the coordinated activity patterns across these areas. While the acquisition rate of the imaging system and the kinetics of the expressed calcium indicator (*Chen et al., 2013b*) used in this study precludes our ability to capture the 4–10 ms spike latencies reported across mouse cortical areas (*Ferezou et al., 2007*) for examining spike-timing and monosynaptic relationship of long-range cortical dynamics, we nevertheless observe that population activity across S1 and S2 is coordinated during relevant task periods in a behavior-dependent manner. We took a simplified view of the population activity by performing dimensionality reduction with LDA, which is a supervised method to

project high-dimensional dynamics onto a single axis. Specifically, through the analysis of correlated population responses across S1 and S2 along multiple LDA axes, we find that inter-areal coordination is associated with both goal-directed licking as well as whisking behavior and that it can occur independent of sensory stimuli. In the absence of tactile stimuli, $S1_{S2}$ and $S2_{S1}$ neurons do not show a special contribution to motor-related coordination, suggesting that this coordination does not necessarily reflect direct cortico-cortical interactions. S1 and S2 receive common input from M1 and POm, conveying efferent and re-afferent motor information (*Deschenes et al., 1998*; *Suter and Shepherd, 2015*), and are additionally coupled by thalamic relays through POm (*Theyel et al., 2010*) (*Figure 7*). We speculate that motor-related S1 and S2 coordination could be a result of common drive or cortico-thalamocortical pathways through these shared areas. In contrast, we find a special contribution of identified $S1_{S2}$ and $S2_{S1}$ neurons to inter-areal coordination occurring during whisker-texture touch, indicating that their participation particularly depends on sensory stimuli. This contribution is most prominent when sensory stimuli and motor behavior are paired, such as upon licking onset on Hit trials, which further suggests that such cortico-cortical interactions could be involved in a form of 'active sensation'. However, we reason that this interaction does not necessarily reflect motor behavior. Given that these neurons also specially contribute to inter-areal coordination during whisker-touch periods in CR trials, we propose that these direct cortico-cortical interactions more likely represent the exchange of sensory- or decision-related activity.

Our findings provide the first direct evidence for a unique contribution of direct cortico-cortical interactions over other sources of correlated activity across these areas. Such specificity points

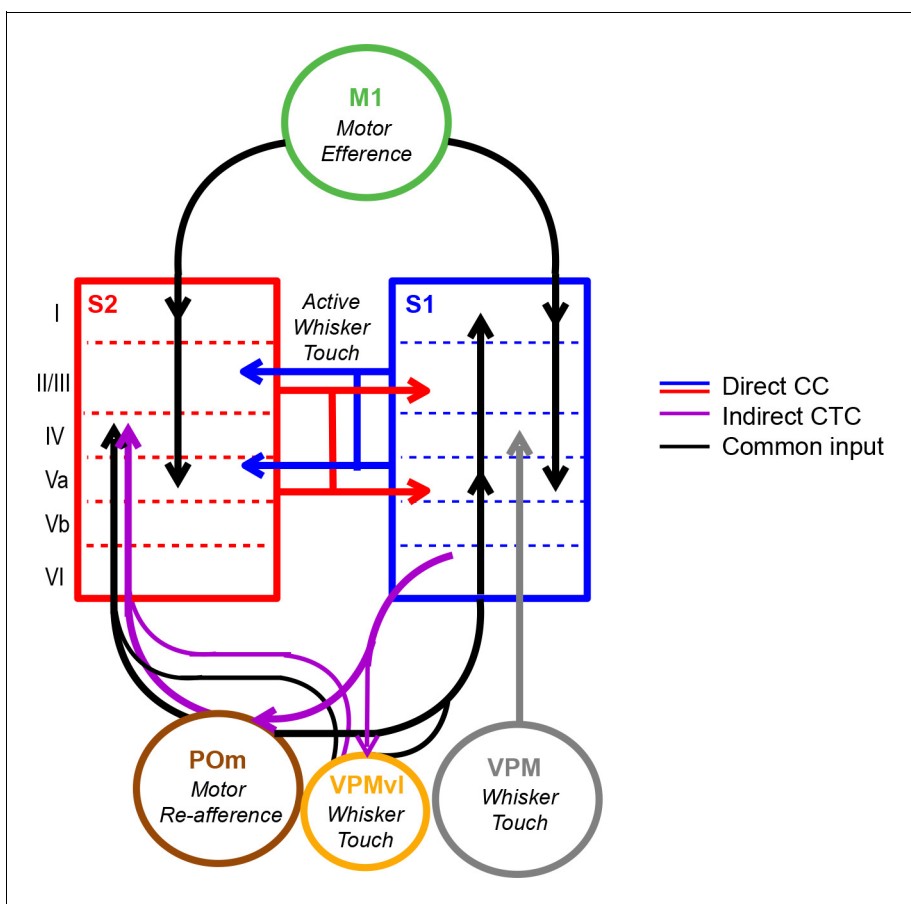

**Figure 7.** Model of coordinated activity across S1 and S2. Our results identify coordinated activity patterns across S1 and S2 that are related to motor behaviors, which could arise from common input from M1 or POm or indirect cortico-thalamocortical (CTC) pathways through POm. $S1_{S2}$ and $S2_{S1}$ neurons especially participate in inter-areal coordination when motor behavior is paired with sensory stimuli suggesting that such cortico-cortical (CC) interactions specifically reflect the exchange of sensory information during active sensation.

towards distinct but potentially synergistic roles for how different inputs may be involved in information flow across the cortex. The division between direct and indirect interactions along sensory and motor parameters, respectively, is in line with theories that indirect cortico-thalamocortical pathways are more involved in relaying motor, rather than sensory, signals (*Sherman and Guillery, 2011*). Additionally, both frontal cortical areas such as M1 and higher-order thalamic nuclei such as POm have been implicated in gating and coordinating activity across somatosensory areas (*Pais-Vieira et al., 2013*; *Theyel et al., 2010*; *Zagha et al., 2013*). In the visual system, the pulvinar, a higher-order thalamic nuclei involved in attention, has been identified in coordinating activity between visual areas (*Saalmann et al., 2012*). Our findings support the notion that nuclei that also drive motor-related or attention-related activity might help to coordinate primary and higher sensory areas in a manner that facilitates sensory-related direct cortico-cortical communication upon stimulus presentation.

What is the relevance of this sensory-related cortico-cortical interaction? It has been suggested that feedback inputs from higher sensory areas provide contextual information to help extract relevant sensory features provided by feedforward inputs in primary areas (*Gilbert and Li, 2013*). We find that $S2_{S1}$ neurons exhibit prominent licking and decision-related activity, which is in line with recent evidence that behavior-related activity in S1 can be inherited from S2 (*Yang et al., 2016*). Coordinated activity between $S1_{S2}$ and $S2_{S1}$ neurons during texture discrimination may reflect sensory processing involved in several functions. One function may be associated with decision making, as exemplified by correlations increasing upon whisker touch onset and peaking upon licking onset on Hit trials. Another function may be associated with the reinforcement of particular aspects of the sensory signal that might strengthen or stabilize sensory representations through goal-directed learning, exemplified by the persistent coordinated activity after licking onset on Hit trials (*Chen et al., 2015*). Future work to dissect $S1_{S2}$ and $S2_{S1}$ neuronal dynamics using projection-targeted multi-area calcium imaging will help to resolve these possibilities.

In conclusion, multi-area calcium imaging with anatomical tracers presents new opportunities for overlaying functional measurements with recent comprehensive mapping of the long-range connectivity in mouse neocortex (*Oh et al., 2014*; *Zingg et al., 2014*). Several different approaches have been implemented for imaging across brain areas. While the use of multiple miniature objectives does not limit the maximum distance between areas that can be imaged (*Lecoq et al., 2014*), the physical size and working distance of such objectives does limit the proximity between areas and depth that can be imaged noninvasively. The use of multiple beams through a single large-FOV, long-working distance objective thus provides a complementary approach. The multi-area two-photon miscroscope described here shares similar design principles as reported in *Stirman et al. (2014)*. The core principles and modularity of these designs readily allows for improvements in temporal resolution through resonant, free-line, or random-access scanning systems (*Bathellier et al., 2012*; *Grewe et al., 2010*), increasing the number of simultaneously imaged areas and cortical layers with low repetition rate lasers (*Cheng et al., 2011*; *Quirin et al., 2014*), and imaging across larger FOVs with different optical configurations (*Stirman et al., 2014*; *Tsai et al., 2015*).

In addition to developments in imaging technology, new genetic tools are being developed for combinatorial conditional gene expression to concurrently label increasing number of pathways (*Fenno et al., 2014*) and genetically encoded voltage indicators for reporting electrical signals (*Akemann et al., 2013*; *Sun et al., 2013*). These developments will expand the range of biological questions that can be addressed in elucidating the relationship between long-range cortical communication and the fine-scale organization and computations occurring within local circuitry.

## Materials and methods

### Multi-area two-photon microscope

The microscope consists of several building blocks: The beam preparation stage, which sends two pulse trains with the correct delay and intensity to the focal plane units (FPUs), which in turn allow independent focusing and positioning of each sub-area. The scan system scans the beams directed to each sub-area in parallel and sends them to the objective via the excitation optics. The microscope front end consists of the objective, z-stage and detection system. FPUs, excitation optics and

the microscope front end were mounted on an elevated breadboard. The optical design software Zemax (Zemax LLC, Redmond, USA) was used for system layout and performance evaluation.

## Beam preparation

Laser light from a Ti:sapphire laser (Mai Tai HP DeepSee, Spectra-Physics) was split into two beams using a 50:50 femtosecond beam splitter (10RQ00UB.4, Newport). In each beam path, a Pockels cell (Model 350-80-LA-02, Conoptics, Danbury, CT) was used to adjust the laser intensity. A 4x beam expander composed of a −25 mm and 100 mm achromat (ACN127-25B and AC254-100B-ML, Thorlabs) was matched to the beam size to the FPU entrance aperture. An adjustable delay line was implemented for one beam path to allow tuning of the relative phase of both excitation beams.

## Focal plane units

The two focal plane units were built in mirrored design from opto-mechanical parts on 200x450 mm breadboards (MB2060/M, Thorlabs) mounted on linear translation stages (Standa 8MTF-102LS05, Vilnius, LT) with kinematic seats (KBS98B, Thorlabs) to allow for quick exchange of FPU configurations. For focusing each FPU contained an electrically tunable lens (Optotune EL-10-30-C, selected for low wavefront aberrations, focal length tuning range: 80 to 230 mm, Optotune AG, Zurich, CH), positioned in a plane conjugate to the scanners and coupled to the first scan lens via a relay lens (Thorlabs AC254-125-B-ML) and a small fold mirror (Thorlabs MRA10-E03). For tuning the beam from convergent to divergent (equivalent to decreasing or increasing the working distance of the objective, respectively), each ETL was combined with a negative offset lens (f = −120 mm, Qioptiq). In each FPU a relay lens (Thorlabs AC254-125-B-ML) and a small fold mirror (Thorlabs MRA10-E03) was used to direct the beam into the scan system.

## Scan system

The output beams from the two FPUs were coupled into the scan system which consisted of two identical scan lenses (S4LFT0089/98, Sill Optics, Wendelstein, D) via the cantilever arms. XY-movement of the FPUs and ETL-focusing enabled independent positioning and focusing of the two imaging sub-areas for the two beams. 6-mm galvanometric scan mirrors were used in the scan unit (6215H, Cambridge Technology, Bedford, MA).

## Excitation optics

A 200 mm tube lens (AC508-200B, Thorlabs) coupled the excitation beams into the microscope objective (16X CFI75 LWD (NA 0.80), Nikon, Egg, CH). With 6-mm scan mirrors, the effective excitation NA was 0.53, which under-filled the microscope objective.

## Microscope front end

The objective and the detector system were mounted on a custom-made Z-translation stage with high load capacity (Feinmess LT235-50-DC-R-B, Dresden, Germany). A crossed-roller ring bearing (THK RU178UUCC0G, Tokyo, Japan) allowed the rotation of the Z-stage, objective and detection system to accommodate for different tilts of the cover slip of the chronic window preparation.

## Detection optics and electronics

The detection system of the microscope was optimized for high detection efficiency over large FOVs by making use of the large acceptance angles of the hybrid photo detectors (HPDs, R11322U-40 MOD, Hamamatsu). To de-magnify the 20 mm-pupil of the microscope objective onto the 5 mm-active area of the HPDs, we used a 4.7x telescope composed of a 90 mm-Achromat (G322389000, Qioptiq) and a wide-angle eyepiece (Panoptic 19 mm, TeleVue). A dichroic mirror (515DCXR, Chroma Technologies) located between the achromat and eyepiece split the emission light into two channels in which blue (480/60 nm, Semrock) and yellow (542/50 nm, Semrock) emission filters and IR rejection filters (FF01-720/SP-25, Semrock) were located. The HPD signal was preamplified (C1077B, Hamamatsu) and digitized by an analog-to-digital converter (ADC, NI-5771, National Instruments) connected to a field-programmable array (FPGA, NI-7962R, National Instruments). For synchronizing the data acquisition to the laser pulse train (*Cheng et al., 2011*), the signal of the internal reference photo diode of the laser was sent to an adjustable electronic delay line (DB64,

Stanford Research Systems) and amplified (BBA100VG, Alphalas) before being fed into the trigger line of the ADC. The FPGA counted the pulses arriving from the two excitation beams and sorted the resulting HPD emission signal accordingly in order to separate the images acquired from two sub-areas at a rate of 160 MHz.

## Microscope software

For microscope control, a custom-written software 'SCOPE' was programmed in C++ (Visual Studio C++) and used to control live scanning, data acquisition, laser intensity, FPU and ETL movement, and the synchronization to other experimental equipment. For combinatorial plane hopping during behavior experiments, custom behavior software programmed in LabVIEW was used to trigger multi-area imaging through SCOPE and control laser intensity and ETL focus shift. Documentation is available at http://rkscope.sourceforge.net/.

## Generation of viral construct

For construction of the *AAV-EF1α-fio-H2BLSSmKate2* viral construct, the double-inverse oriented FRT (*fio*) sites was synthesized de novo (GenScript) with flanking *Bam*H1 and *Eco*RI restriction sites and an internal *Asc*I and *Nhe*I sites and insert into an *AAV-EF1α-YC-Nano140* (*Chen et al., 2013b*) backbone plasmid. The *H2B* subunit with 5' *Nhe*I and 3' *Age*I restriction sites was generated by PCR amplification from a *pTagRFP-H2B* vector (Evrogen) and subcloned into an *pLSSmKate2-N1* plasmid (*Piatkevich et al., 2010*). Subsequently, *H2BLSSmKate2* with 5' *Nhe*I and 3' *Asc*I restriction sites was generated by PCR amplification followed by insertion into the *AAV-EF1α-fio* plasmid. The *AAV-syn-Flpe* viral construct was generated by restriction enzyme digest of *pCAG-Flpe* (*Matsuda and Cepko, 2007*) and insertion into the *pAAV-6P-SEWB* backbone plasmid. For the *AAV-EF1α-dio-NLStdTomato* viral construct, *NLStdTomato* with 5' *Nhe*I and 3' *Age*I restriction sites was generated by PCR amplification from a *pTagRFP-H2B* vector followed by insertion into the *AAV-EF1α-dio-eYFP* plasmid. The *AAV-pgk-Cre* construct was previously described[3]. Recombinant serotype 6 AAV particles were produced by co-transfecting AAV-293 cells with the shuttle plasmid and the pDP6 packaging plasmid. Recombinant serotype 1 AAV particles were produced by co-transfecting AAV-293 cells with the shuttle plasmid and the pDF1 packaging plasmid. Cell lysates were subjected to purification on iodixanol density gradients followed by HPLC with HiTrap Heparin column for AAV2 or by anion exchange HPLC for AAV1 (GE Healthcare Bio-Sciences AB) using standard procedures. The viral suspension obtained was concentrated using Centricon centrifugal filter devices with a molecular weight cut-off of 100 kDa (Millipore, Billerica, MA), and the suspension medium replaced with PBS. Vector titres were determined by measuring the number of encapsidated genomes per ml by real-time PCR.

## Viral injections and cranial window implantation

Experimental procedures followed the guidelines of the Veterinary Office of Switzerland and were approved by the Cantonal Veterinary Office in Zurich. Stereotaxic viral and tracer injections were performed on young adult (P35-42) male wild type C57Bl6/J mice as previously described (*Chen et al., 2013b*). A solution containing AAV1-*EF1α-YC-Nano140*, AAV1-*EF1α-fio-H2BLSSmKate2*, and AAV6-*pgk-Cre* (200 nl total volume, ~1 x 10$^9$ vg/µl per virus, 2:1:1 ratio by volume) was delivered into S1, targeting L2/3 and L5 (~300 and 500 µm below the pial surface). A solution containing AAV1- *EF1α-YC-Nano140,* AAV1-*EF1α-dio-NLStdTomato,* and AAV6-*syn-Flpe* (200 nl total volume, ~1 x 10$^9$ vg/µl per virus, 2:1:1 ratio by volume) was delivered into S2, targeting L2/3 and L5 (~300 µm and 500 µm below the pial surface). Injection regions were selected by optical intrinsic signal imaging or stereotaxic coordinates (1.1 mm posterior to bregma, 3.3 mm lateral for S1; 0.7 mm posterior to bregma, 4.2 mm lateral for S2). To allow long-term in vivo calcium imaging, a cranial window was implanted 24 hr after virus injections over S1 as described[33]. A metal post for head fixation was implanted on the skull, contralateral to the cranial window, using dental acrylic. For demonstration of large single-FOV imaging, structural images were acquired from one adult male Rasgrf2-2A-dCre;Camk2atTA;Ai92(TITL-YCX2.60) transgenic mouse (*Madisen et al., 2015*) implanted with a cranial window without viral injections.

## Animal behavior

Mice were housed 2–3 per cage in reverse 12 hr light cycle conditions. All handling and behaviour occurred under simulated night time conditions. One week following chronic window implantation, mice were handled daily for 1 week while acclimated to a minimum of 15 min of head fixation. Mice were water restricted and trained to a go/no-go texture discrimination task previously described (*Chen et al., 2013b*). Imaging during behaviour began following 3–5 training sessions once animals reached a performance level of d' > 1.75 (80% correct) for one session. Imaging under task conditions was performed over the course of 5–6 sessions at a performance level of d' = 2.62 ± 0.15. Once sufficient task-related data was acquired, mice were provided with free access to water and then imaged for an additional 5–6 sessions under non-task conditions, in which textures were presented but no reward or punishment delivered. Sample sizes were chosen based on previous behavioural imaging studies, which comprise 6–10 mice per group (*Chen et al., 2013b*; *2015*). Due to their low occurrence (6.7 ± 0.5% of all trials), miss trials were excluded from analysis. No statistical methods were used to predetermine sample size. Since animals constitute a single experimental group, no randomization of animals or blinding to experimenter was performed.

## Intrinsic signal optical imaging

The S1 and S2 barrel column was identified using intrinsic signal optical imaging under ~1.5% isoflurane anaesthesia. The cortical surface was illuminated with 630-nm LED light, single whiskers were stimulated (2–4° rostro-caudal deflections at 10 Hz), and reflectance images were collected through a 4x objective with a CCD camera (Toshiba TELI CS3960DCL; 12-bit; 3-pixel binning, 427x347 binned pixels, 8.6-μm pixel size, 10-Hz frame rate). Intrinsic signal changes were computed as fractional changes in reflectance relative to the pre-stimulus average (50 frames; expressed as $\Delta R/R_{IOS}$). Barrel column centres for stimulated whiskers were located by averaging intrinsic signals (15 trials), median-filtering (5-pixel radius), and thresholding to find signal minima. Reference surface vasculature images were obtained using 546-nm LED illumination and matched to images acquired during 2-photon imaging. Prior to behavior training, all whiskers excluding the principal and first-order surround whiskers corresponding to the expression area were partially trimmed to a length out of reach from texture contact during the task. During whisker trimming, the principal whisker was noted by images taken from the high-speed video camera for re-identification in subsequent imaging sessions for whisker tracking.

## Whisker tracking

The whisker field was illuminated with 940-nm infrared LED light and movies were acquired at 500 Hz (500 x 500 pixels) using a high-speed CMOS camera (A504k; Basler). Average whisker angle across all imaged whiskers was measured using automated whisker tracking software (*Knutsen et al., 2005*). Because our limited temporal resolution of imaging (7 Hz) precluded analysis of rapid dynamics within individual whisking cycles, we based our analysis on the envelope amplitude of whisking calculated as the difference in maximum and minimum whisker angle along a sliding window equal to the imaging frame duration (142 ms). The slower dynamics of the envelope amplitude represents both rhythmic and non-rhythmic forms of whisking behavior. For comparison between high- vs. low-amplitude whisking trials, the mean whisking amplitude during the 1-s period prior to whisker-touch onset was calculated for each animal and high- and low-amplitude trials were identified as those whose amplitude during the same period was greater or less than the mean, respectively. For all trials, the first and last possible time point for whisker to texture contact was quantified manually through visual inspection.

## Identification of feedforward and feedback neurons

For in vivo identification of LSSmKate-positive feedforward and tdTomato-positive feedback neurons, 3D-volume image stacks were taken on a standard custom-built 2-photon microscope controlled by HelioScan[34], equipped with a Ti:sapphire laser system (~100-fs laser pulses; Mai Tai HP; Newport Spectra Physics), a water-immersion objective (40×LUMPlanFl/IR, 0.8 NA; Olympus), galvanometric scan mirrors (model 6210; Cambridge Technology), and a Pockels Cell (Conoptics) for laser intensity modulation. An 800-nm excitation with 610/75 nm emission filter and 840–900 nm excitation with 697/75 nm emission filter was used for tdTomato and LSSmKate2, respectively.

Due to suboptimal in vivo 2-photon excitation of LSSmKate2, additional H2BLSSmKate2-positive neurons were identified followed by antibody staining of LSSmKate2 for signal amplification. Mice were anesthetized (ketamine/xylazine; 100/20 mg/kg body weight) and perfused transcardially with 4% paraformaldehyde in phosphate buffer, pH 7.4. Cortical sections (50 µm) were cut along the imaging plane using a vibratome (VT100; Leica), then blocked in 10% NGS and 1% Triton at room temperature and incubated overnight at 4°C in 5% NGS, 0.1% Triton and mKate guinea pig poly-clonal antibody (*Cai et al., 2013*); 1:1,000). A guinea pig Alexa647–conjugated goat IgG secondary antibodies (1:400; Molecular Probes, Invitrogen) was applied for 2 hr at room temperature. Images were acquired with a confocal microscope (Fluoview 1000; Olympus), green (YC-Nano140), red (tdTomato), and infrared (Alexa647) excitation/emission filters.

### Calcium imaging analysis

Two-channel, two-area (CFP/YFP) calcium imaging data was imported into MATLAB (Mathworks) for processing. For each channel, spatial linear unmixing was applied for the two area as described below. Background was subtracted on each area and channel (bottom $1^{st}$ percentile fluorescence signal across entire frame). For each area, Hidden Markov Model line-by-line motion correction was applied to both data channels. Regions of interests (ROIs) corresponding to individual neurons were manually selected from the mean image of a single-trial time series using ImageJ (National Institute of Health). Mean pixel value for each ROI was extracted for both channels. Calcium signals were expressed as relative YFP/CFP ratio change $\Delta R/R=(R-R_0)/R_0$. $R_0$ was calculated for each trial as the bottom $8^{th}$ percentile of the ratio for the trial. Active neurons were identified by two-way ANOVA with repeated measures of the neuronal calcium signal against the neuropil signal (significance value, $p<0.05$) for each imaging session. The neuropil is defined as a region of interested selected from the entire imaging frame representing non-somatic tissue expressing YC-Nano140 but excluding blood vessels.

### Denoising with tensor decomposition

Calcium signals were denoised using tensor decomposition before further analysis (*Figure 2— figure supplement 1* and *Figure 2—source data 1*) (*Cong et al., 2015*; *Seely et al., 2014*). Tensor decomposition is a method used for dimensionality reduction, which can be viewed as a generalization of singular value decomposition of data represented as tensors rather than matrices (*Hitchcock, 1927*). While calcium imaging recordings are often described as two-dimensional matrices comprised of neurons and time dimensions, it can additionally be described along a third dimension representing trial conditions (*Figure 2—figure supplement 1A*). For such data, tensor decomposition can be used as a form of single-trial denoising by assuming that calcium signals across neurons, time, and trial conditions are not independent and that multi-linear relationships across dimensions therefore can be exploited. Through tensor decomposition, background noise that does not match the assumed multi-linear structure can be reduced if present. Single-trial denoising of calcium transients is desirable when analyzing conditions with low trial counts such as FA trials (7.4% of all trials) in order to improve statistical analysis of such conditions.

For each animal, calcium signals were arranged into a data tensor (Y) across three dimensions according to the number of trial conditions (*I*; i.e., 6 combinations of decision and texture), number of neurons (*J*), number of time points (*K*). Using Tucker decomposition, this tensor can be described elementwise as:

$$y_{ijk} = \sum_{c=1}^{C} \sum_{n=1}^{N} \sum_{t=1}^{T} g_{cnt} m_{ic} m_{jn} m_{kt} \quad \text{for } i=1...I, \, j=1...J, \, k=1...K \tag{1}$$

consisting of a factor matrix related to trial-condition containing elements ($m_{ic}$) with column size C, a factor matrix related to neuron containing elements ($m_{jn}$) with column size N, a factor matrix related to time point ($m_{kt}$) with column size T, and a core tensor describing the interactions between the matrix components containing elements ($g_{cnt}$). From this, a low rank tensor, Y′, containing the denoised traces can be described elementwise as:

$$y'_{ijk} = \sum_{c=1}^{C'} \sum_{n=1}^{N'} \sum_{t=1}^{T'} g_{cnt} m_{ic} m_{jn} m_{kt} \quad \text{for } i = 1...I, \ j = 1...J, \ k = 1...K \tag{2}$$

This tensor is obtained by reducing the column size of each factor matrices across each dimension resulting in $C'$ which is related to the number of trial conditions such that ($C' \leq I$), $N'$ which is related to the number of neurons such that ($N' \leq J$), and $T'$ which is related to the number of time points such that ($T' \leq K$). From this, a tensor rank ($\Theta'$) for $Y'$ can be expressed as the sum of the reduced column sizes across all dimensions:

$$\Theta' = C' + N' + T' \tag{3}$$

In order to determine the optimum $\Theta'$, a five fold cross validation procedure was first performed (*Figure 2—figure supplement 1B*) (*Seely et al., 2014*). For each trial condition in each neuron, trials were divided into a training set (80% of trials) and a test set (20% of trials). Single-trial traces in each tensor element were replaced with average traces from the training set. Denoised traces were obtained for a given $\Theta'$ and compared to the average traces of the test set by computing the mean squared errors (*MSE*) (*Figure 2—figure supplement 1C,D*). The optimum $\Theta'$ is identified as $\Theta'$ with the minimum *MSE*. Determining $\Theta'$ by five fold cross validation is advantageous in that it is unsupervised and can correct for unknown sources of noise. However, since the error estimation used in this procedure is based on comparing average traces, the $\Theta'$ determined is not necessarily optimized for denoising single-trial responses and thus neurons with variable trial-to-trial responses may not be properly denoised. Indeed, while five fold cross validation was sufficient in identifying optimum $\Theta'$ for $T'$ and $C'$, better fits for some neurons were observed when manually adjusting $N'$ (data not shown).

In order to improve denoising of single-trial responses, a second-step procedure was implemented to optimize $N'$ through a supervised approach of performing tensor decomposition on noisy simulated calcium transients in order to determine a rank offset ($N'_{offset}$) resulting in a final tensor rank $\left(\Theta'_{final}\right)$ such that:

$$\Theta'_{final} = C' + N' + N'_{offset} + T' \tag{4}$$

where the denoised transient best reflects the ideal transients.

In order to emulate the multi-linear structure across neurons, time, and trial conditions present in our experimental data that is required for tensor decomposition, a peeling algorithm (*Grewe et al., 2010*) using previously measured YC-Nano140 parameters (*Chen et al., 2013a*) (single-action potential transient: $A_0$ = 4.54%, $\tau_{onset}$ = 0.186 s, $A_{peak}$ = 2.3%, $\tau_{decay}$ = 0.673 s) was applied to raw traces to extract estimated spike trains for all neurons and trials for a single animal. While the accuracy and precision of the estimated spikes may vary depending on noise in the raw trace (*Lütcke et al., 2013*), the multi-linear relationships across each tensor dimension is preserved. The estimated spike trains are then convolved using YC-Nano140 parameters to produce ideal simulated calcium transients. The degree of noise under experimental conditions is estimated by assuming that any variance in calcium signal present in inactive neurons reflects non-neuronal noise. For each inactive neuron, a normal distribution was fit to raw calcium traces to obtain $\sigma$ representing the degree of noise for that neuron. Noise was then added neuron-by-neuron to simulated calcium transients that matched the $\sigma$'s from all inactive neurons in the data set.

The similarity between the ideal and denoised simulated trace was measured by computing the Pearson's correlation coefficient (*CC*) between the two traces for each neuron and taking the average across neurons. From this, the optimum $\Theta'_{final}$ was determined by calculating a cost function representing the difference between $\Theta'_{final}$ and the CC obtained from $\Theta'_{final}$, each normalized across the range of tested $\Theta'_{final}$:

$$Cost\left(\Theta'_{final}\right) = \left\| \Theta'_{final} \right\| - \left\| CC\left(\Theta'_{final}\right) \right\| \tag{5}$$

such that the optimum $\Theta'_{final}$ resulted in the minimum $Cost(\Theta'_{final})$ (*Figure 2—figure supplement 1E*).

In comparing denoising of simulated transients with tensor decomposition against temporal smoothing by a 5-point Gaussian filter, we observed that denoising with tensor decomposition better preserves the onset and peak of calcium transients, resulting in better *CC* of denoised to ideal traces (Tensor decomposition $0.69 \pm 0.01$, Gaussian filter: $0.65 \pm 0.01$, $p<1\times10^{-6}$, Student's *t*-test, *Figure 2—figure supplement 1F,G*). This suggests that denoising with tensor decomposition is preferred when investigating sub-second temporal dynamics of activity as it preserves high frequency components of the calcium signal.

The optimum $\Theta'_{final}$ for each animal was determined for denoising (*Figure 2—figure supplement 1H*). We asked if the size of the optimum low rank tensor used for denoising was similar across animals (*Figure 2—source data 1*). We observed that *C'* was largely consistent across animals and reflected a rank near the total possible ranks along the condition dimension. For *T'* and $N' + N'_{offset}$, we observed that the optimum column size across these dimensions was strongly correlated with the number of identified active neurons (*T'*: $R = 0.82$, $p<0.05$; $N' + N'_{offset}$: $R = 0.82$, $p<0.05$, Pearson's correlation, *Figure 2—figure supplement 1I*). This suggests that the optimum low rank tensor identified for denoising captures a relevant portion of the original data tensor containing real calcium transient events.

## Spatial linear unmixing

Spatial linear unmixing is based on the fact that the total PMT signal recorded at the corresponding pixel for both areas in a given channel is the linear sum of the signal for each area weighted by the cross talk resulting from the fluorescence lifetime of the indicator. For a dual beam system, the contribution of the two detected areas can be represented by the following equations:

$$J_1(x,y) = S_{1,1} \times I_1(x,y) + S_{1,2} \times I_2(x,y)$$
$$J_2(x,y) = S_{2,1} \times I_1(x,y) + S_{2,2} \times I_2(x,y)$$

(6)

where *J* is the total signal per area, *I* is the fluorophore abundance, and S is the crosstalk. These equations can be expressed as a matrix:

$$[J] = [S][I]$$

(7)

whereby the unmixed image [*I*] can be calculated using the inverse matrix of [*S*]

$$[I] = [S]^{-1}[J]$$

(8)

assuming the detected signal in both areas represents the total signal:

$$S_{1,1} + S_{2,1} = 1$$
$$S_{1,2} + S_{2,2} = 1$$

(9)

[S] [*S*] was determined empirically at the beginning of each session using the experimentally prepared mouse expressing YC-Nano140. The intended FOVs were sequentially scanned with a single excitation beam during dual area acquisition mode. The resulting crosstalk into each area was calculated from the acquired reference images and applied for spatial linear unmixing of subsequent dual beam data using MATLAB.

## Behavior classification

Behavior-related activity was described using a general linear model (GLM) (*Miri et al., 2011*; *Pinto and Dan, 2015*) expressed as:

$$Y_t = \sum_{i=-3}^{6} B_i^L X_{t-i}^L + \sum_{i=-3}^{6} B_i^W X_{t-i}^W + \sum_{i=-3}^{6} B_i^T X_{t-i}^T$$

(10)

Z-scored regressors ($X_{t+i}$) representing touch onset (*T*), whisking envelope amplitude (*W*), and licking onset (*L*) with regression coefficients ($B_i$) at different delays (*i*) were used to model the z-

scored calcium signal ($Y_t$) across time frames $t$. Regressors for touch onset and whisking amplitude were obtained from the whisker-tracking video while regressors for licking onset were obtained from the lick-sensor data. Each regressor was down sampled to match the calcium imaging frame rate. Touch onset was selected to best reflect touch-related responses given previously reported neuronal adaptation in neuronal firing upon repeated touches (*Musall et al., 2014*; *Yamashita et al., 2013*). Whisking envelope amplitude was previously observed to best reflect periods of whisking and non-whisking behavior in order to identify whisking-related neurons (*Chen et al., 2013b*). Given the slow kinetics of calcium indicators and given that the imaging rate is well below the Nyquist rate of the natural whisking frequency (~10 Hz) (*Kleinfeld and Deschenes, 2011*), whisking-related signals measured here do not reflect whisking frequency. Licking onset was selected due to the observation that licking behavior in task-performing mice typically proceeds in licking bouts. Introducing additional behavioral regressors such as licking offset and touch offset to the GLM did not improve model fit (data not shown). In order to capture a physiologically realistic range of response delays to behavioral events as previously observed (*Chen et al., 2013b*; *2015*), regressors for each behavioral parameter were generated across a range of delays from $i = -3$ ($t = 0.43$ s before behavior event) to $i = +6$ ($t = 0.85$ s after behavioral event). Only delays from $i = -2$ to $i = +5$ were included for cell type analysis. GLM was applied to active neurons, where the first 5 s from each trial across active sessions were extracted and concatenated for analysis.

To fit the GLM, trials were randomly divided into a training set (80% of trials) and a test set (20% of trials). Ridge regression was used to minimize the $B_i$ at irrelevant delays. The optimum regularization parameter was determined by performing a five fold cross validation within the training set and selecting the value with the best cross validation performance. $B_i$ was then calculated from the training set and applied to the test set to obtain predictions for $Y_t$. To assess GLM fit, a coefficient of determination ($R^2$) was calculated by comparing the predicted and the original traces. To reduce the effect of the particular choice of test trials on $R^2$, test trials were bootstrapped 1000 times to obtain a final $R^2$ reflecting GLM fit. To assess the significance of individual $B_i$, a shuffled distribution for each $B_i$ was obtained by permutation test after shuffling calcium traces for time points within each trial 1000 times. $B_i$ whose value was greater than the 95 percentile of the shuffled distribution was identified as significant. GLM does not require normal distribution of the data set. Comparisons of $B_i$ across cell types was performed using one-way repeated measures ANOVA. The variances of each cell type were tested using the F-test and determined to not be significantly different.

## Trial type analysis

The performance of neuronal populations or single neurons in discriminating two trial types was assessed using a receiver operating characteristic (ROC) analysis (*Green and Swets, 1966*; *O'Connor et al., 2010*). For neuronal populations, the discriminability of the population response projected along the LD axis was measured at each time point 1 s prior to and following touch. Each trial was assigned a 'discrimination variable' score (DV) equal to the similarity to the mean projected population response for trial type $X$ minus the similarity to the mean projected population response for trial type $Y$. Thus, for trial type $X$

$$DV_x = X_i\left(\bar{X}_{\forall j \neq i} - \bar{Y}\right) \tag{11}$$

and for trial type $Y$

$$DV_Y = Y_i\left(\bar{X} - \bar{Y}_{\forall j \neq i}\right) \tag{12}$$

where $X_i$ and $Y_i$ are the single-trial population response for the $i$-th trial. $\bar{X}$ and $\bar{Y}$ are the mean population response. Trials were classified as belonging to trial type $X$ or $Y$ if $DV_X$ or $DV_Y$ was greater than a given criterion, respectively. To determine the fraction of trials an ideal observer could correctly classify, an ROC curve was constructed by varying this criterion value across the entire range of $DV_X$ or $DV_Y$. At each criterion value, the probability that a trial of type $X$ exceeded the criterion value was plotted against the probability that a trial of type $Y$ exceeded the criterion value. The area under the ROC curve ($A_{observed}$) was then calculated to represent the single-neuron performance ('fraction correct') as the fraction of trials correctly discriminated by an ideal observer using the DV. We corrected for sampling bias due to the limited number of trials collected, using methods described (*Safaai et al., 2013*). The sampling bias ($A_{bias}$) was determined by calculating the mean

area under the ROC curve after randomly shuffling trial type or stimulus labels repeated 1000 times. The corrected area under the ROC curve $A_{corrected}$ was then calculated as $A_{corrected} = A_{observed} - A_{bias} + 0.5$.

For single neurons, classification of trial type $X$ versus trial type $Y$ was based on the similarity of the calcium transient in each trial to the mean calcium transient for trial type $X$ compared to trial type $Y$. Only the first second of the calcium signals following initial texture contact was considered since it reflected the minimum touch duration common across trial types (*Chen et al., 2013b*; *2015*). $DV$ was equal to the dot-product similarity to the mean calcium transient for trial type $X$ minus the dot-product similarity to the mean for trial type $Y$. Neurons discriminating above chance were identified using repeated permutations tests where trial type or stimulus labels were randomly shuffled. For each permutation test, a threshold corresponding to the shuffled distribution 95[th] percentile was calculated. Neurons, whose performance values were above the mean value of this threshold across 1000 permutation tests, were considered to be discriminating above chance. Comparison of discriminative neurons across cell types was performed using a $\chi^2$ test. Normal distribution was assumed for statistical comparison but not explicitly tested.

## Linear discriminant analysis

We used linear discriminant analysis (LDA) for dimensionality reduction of neuronal population responses. Observations consisted of the $\triangle R/R$ values at a given time point for all neurons simultaneously recorded within an imaging field, thus representing the neuronal state space vector at this moment (with each neuron representing one dimension), i.e., representing a 'snapshot' of the state space vector trajectory during the given trial. Observations were considered for all $n$ trials, separated into the $N_1$ and $N_2$ trials for the two chosen trial conditions $C_1$ and $C_2$, respectively (e.g., Hit vs. CR or low- vs. high-amplitude whisking; see *Table 1*). $\triangle R/R$ values were arranged in a matrix **x** with neurons as columns and trials as rows. The LDA procedure seeks to find a projection vector $w$ such that the projections of the observations onto this axis, collected in the vector

$$y = w^T x + w_0, \tag{13}$$

are best separated for the two chosen trial conditions. Maximal separation is defined as the maximal difference of the mean vectors $\mu_1 = \frac{1}{N_1} \sum_{n \in C_1} x_n$ and $\mu_2 = \frac{1}{N_2} \sum_{n \in C_2} x_n$ for $C_1$ and $C_2$, respectively, normalized by the within-class scatter. The solution, known as Fisher's linear discriminant (*Fisher, 1936*; *Safaai et al., 2013*), is given by

$$w^T = S_w^{-1}(\mu_1 - \mu_2) \tag{14}$$

where $S_W^{-1}$ is the within-class covariance given by

$$S_W^{-1} = \sum_{n \in C_1} (x_n - \mu_1)(x_n - \mu_1)^T + \sum_{n \in C_2} (x_n - \mu_2)(x_n - \mu)^T \tag{15}$$

The bias is calculated as

$$w_0 = -\frac{1}{2}\left(w^T \mu_1 - w^T \mu_2\right) \tag{16}$$

Intuitively, this procedure finds the hyperplane in the state space (orthonormal to the projection vector $w$ and encompassing $w_0$) that results in best separation according to Fisher's criterion.

To analyze the time courses of neuronal population dynamics during behavior trials, the LDA procedure was applied independently to each time point over 1-s periods before and after whisker-touch onset (or licking onset in some cases). Only neurons identified as active in at least one imaging session were included in the LDA. For each individual trial we thereby obtained a time-dependent 'linear discriminant' variable LD($t$). The mean value $\bar{LD}$ by definition is half of the distance between the projections of the mean vectors $\mu_1$ and $\mu_2$

$$\bar{LD}(t) = \frac{1}{2}\left(w^T \mu_1 + w^T \mu_2\right) \tag{17}$$

For whole-region analysis (S1 or S2) we averaged LD values obtained from all imaging areas/planes.

Inter-areal coordination as a function of time, termed $LDCC_{S1:S2}$, was determined by calculating the Pearson's correlation coefficient between the population responses $LD_{S1}$ and $LD_{S2}$ for S1 and S2, respectively, across all simultaneously imaged trials at each time point. To determine the specific contribution of $S1_{S2}$ or $S2_{S1}$ neurons to cross-areal coordination, a one-dimensional modified discriminant $LD'(t)$ was obtained for each area by shuffling the trial-by-trial calcium responses of $S1_{S2}$ or $S2_{S1}$ neurons, respectively, and then projecting the population vector onto the LDA axis determined from the non-shuffled population response. Cross-correlation of $LD'_{S1}$ and $LD'_{S2}$ yielded $LDCC'_{S1:S2}$. Shuffling was repeated 1000 times to obtain mean and standard error for $LDCC'_{S1:S2}$ values. The change in S1-S2 correlation ($\Delta LDCC_{S1:S2}$) was calculated as the mean $LDCC'_{S1:S2}$ minus the unshuffled $LDCC_{S1:S2}$. Reductions in correlation strength thus show up as negative values. To control for trial shuffling of $S1_{S2}$ or $S2_{S1}$ neurons, trial shuffling was performed on an equal number of $S1_{ND}$ or $S2_{ND}$ neurons, repeated 1000 times, and $\Delta LDCC_{S1:S2}$ was calculated from the average cross-correlation.

Comparisons of $LDCC_{S1:S2}$ and $\Delta LDCC_{S1:S2}$ across trial conditions were performed using one-way repeated measures ANOVA. The variances of each the trial condition were tested using the F-test and determined to not be significantly different.

## Acknowledgements

We thank S Giger, H Kasper, and M Wieckhorst for custom components, B Schneider for viral reagents, P Bethge, D Langer, V Mante, A Thiele, and M van 't Hoff for helpful discussions, J Sanes for donating mKate antibody, H Zeng and A Miyawaki for the Ai92(TITL-YCX2.60) mouse, ME Diamond, M Maravall, L Pinto, and H Safaai for help with data analysis, and F Collman for motion correction software. This work was supported by grants from the Swiss National Science Foundation (310030-127091; FH), the Swiss SystemsX.ch initiative (project 2008/2011-Neurochoice, FH), the US NIH BRAIN Initiative (1U01NS090475-01, FH), Forschungskredit of the University of Zurich (grant 541541808, JLC) and a fellowship from the US National Science Foundation, International Research Fellowship Program (grant 1158914, JLC).

## Additional information

### Funding

| Funder | Grant reference number | Author |
| --- | --- | --- |
| Schweizerischer Nationalfonds zur Förderung der Wissenschaftlichen Forschung | 310030-127091 | Fritjof Helmchen |
| Swiss SystemsX.ch Initiative | 2008/2011-Neurochoice | Fritjof Helmchen |
| National Institutes of Health | BRAIN Initiative 1U01NS090475-01 | Fritjof Helmchen |
| University of Zurich | Forschungskredit grant 541541808 | Jerry L Chen |
| National Science Foundation | International Research Fellowship grant 1158914 | Jerry L Chen |

The funders had no role in study design, data collection and interpretation, or the decision to submit the work for publication.

### Author contributions

JLC, Devised the study, Designed and built the multi-area two-photon microscope, Designed viral constructs and performed in vivo experiments, Wrote the manuscript, Conception and design, Acquisition of data, Analysis and interpretation of data, Drafting or revising the article; FFV, Devised the study, Designed and built the multi-area two-photon microscope, Wrote the manuscript, Conception and design, Drafting or revising the article; MJ, Critical revisions, Analysis and interpretation of data, Drafting or revising the article; RK, Designed and built the multi-area two-photon microscope, Critical revisions, Conception and design, Drafting or revising the article, Contributed

unpublished essential data or reagents; FH, Devised the study, Wrote the manuscript, Conception and design, Analysis and interpretation of data, Drafting or revising the article

## Author ORCIDs

Fritjof Helmchen, http://orcid.org/0000-0002-8867-9569

## Ethics

Animal experimentation: Experimental procedures followed the guidelines of the Veterinary Office of Switzerland and were approved by the Cantonal Veterinary Office in Zurich. Experiments were carried out under the approved licenses 62/2011 and 285/2014.

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
