## [Decision Letter]

Thank you for submitting your article "Long-range population dynamics of anatomically defined neocortical networks" for consideration by *eLife*. Your article has been favorably evaluated by David Van Essen (Senior editor) and three reviewers, one of whom, Andrew King, is a member of our Board of Reviewing Editors. One of the other two reviewers has agreed to reveal his identity: Carlos Portera-Cailliau.

The reviewers have discussed the reviews with one another and the Reviewing Editor has drafted this decision to help you prepare a revised submission.

Summary:

This paper describes the use of a dual-area two-photon microscope for simultaneous imaging of ensembles of anatomically-defined projection neurons in cortical areas S1 and S2 in awake-behaving mice. The authors focused on upper-layer neurons that project from S1 to S2 as well as neurons in S2 that project back to S1, and measured how the relationship between the activity of these neuronal ensembles changes during tactile whisking behavior.

Essential revisions:

The reviewers are enthusiastic about your approach for combining two-photon imaging at multiple z-planes with an elegant retrograde viral labeling approach for targeting anatomically defined subsets of neurons in different cortical areas. They also feel that the results described, although relatively limited in scope, provide a valuable advance over other studies to have used two-photon imaging or whole-cell recording to measure responses of S1 cortical projection neurons. They have, however, identified several concerns, mainly relating to some of the data analysis.

A weakness of the approach used is that imaging was performed at 7 Hz, so that signals only report anticipation or slow activation of neurons as opposed to rapid activation upon touch. Although this is less of an issue for the relatively low activity of L2/3 neurons than for neurons in other cortical layers, this limitation should be mentioned. Similarly, it is not possible to distinguish activity that is modulated by whisking per se, since the Nyquist frequency (3.5 Hz) is much less than the whisking frequency of ~ 10 Hz (see, e.g., Kleinfeld and Deschenes (Neuron 2011) for a review of time-scales). It is critical to state that slow aspects of touch signals only were measured and that the whisking derived signals show slow features – i.e., changes in amplitude and set-point. This is probably best done in the Introduction, in conjunction with a description of the different time scales of whisking. As a technical point, you should test if whisking-based signals were aliased to lie below the Nyquist frequency as this could confound their analysis.

The key part of the paper is the correlation of activity across S1 and S2 as a function of behavioral response. Because the sampling is so slow, this is performed in terms of a population response. The reviewers felt that the linear discriminant analysis was not well explained. Please describe the algorithm used, preferably including the equations on which this was based. For example, was Fisher's linear discriminant derived from the centroids of the responses (with each time point as a dimension) for the hit and correct rejection trials?

Is Figure 5 the only aspect of the paper that truly takes advantage of simultaneous imaging of neurons in both regions? The authors should acknowledge this limitation of their experimental design (namely, that in order to maximize the number of neurons imaged through plane hopping, only 10% of the trials allowed them to analyze recordings from the same cohorts of neurons in both regions).

There are other areas where the paper is difficult to follow, which could limit its potential impact. We therefore recommend that you go over the text carefully, to ensure that readers don't have to guess what was done. In particular, it is essential to adequately explain the technical aspects of the various analyses used for the general audience that *eLife* seeks to reach.

Although the reviewers are not asking for new experiments to be carried out, can you comment on whether the discriminative power of S2S1 neurons changes as the mice learn the task?

Your proposal that S2 projection neurons provide S1 with specific contextual information was felt to be too speculative in the absence of some manipulation showing that disrupting the activity of these cells affects task performance. The Discussion should be rephrased accordingly.

Finally, the reviewers felt that greater acknowledgement should have been made to similar systems that have been developed in several other labs (Lecoq et al., 2014, Nature Neurosci; Stirman et al., 2014, bioRxiv; Tsai et al., 2015). In particular, the study by Stirman et al. is not cited.

---

## [Author Response]

Essential revisions:

The reviewers are enthusiastic about your approach for combining two-photon imaging at multiple z-planes with an elegant retrograde viral labeling approach for targeting anatomically defined subsets of neurons in different cortical areas. They also feel that the results described, although relatively limited in scope, provide a valuable advance over other studies to have used two-photon imaging or whole-cell recording to measure responses of S1 cortical projection neurons. They have, however, identified several concerns, mainly relating to some of the data analysis.

We thank the reviewers for their enthusiasm and overall positive assessment of the study and manuscript. We appreciate the reviewers’ concerns and have addressed them in full, as detailed below.

*A weakness of the approach used is that imaging was performed at 7 Hz, so that signals only report anticipation or slow activation of neurons as opposed to rapid activation upon touch. Although this is less of an issue for the relatively low activity of L2/3 neurons than for neurons in other cortical layers, this limitation should be mentioned. Similarly, it is not possible to distinguish activity that is modulated by whisking per se, since the Nyquist frequency (3.5 Hz) is much less than the whisking frequency of ~ 10 Hz (see, e.g., Kleinfeld and Deschenes (Neuron 2011) for a review of time-scales). It is critical to state that slow aspects of touch signals only were measured and that the whisking derived signals show slow features – i.e., changes in amplitude and set-point. This is probably best done in the Introduction, in conjunction with a description of the different time scales of whisking. As a technical point, you should test if whisking-based signals were aliased to lie below the Nyquist frequency as this could confound their analysis.*

The reviewers are correct that, with our imaging rate of 7 Hz, we could not resolve fast temporal patterns of spiking dynamics on a rapid time scale of tens of milliseconds. All our analysis was therefore done using a downsampled (smoothed) version of ‘whisking amplitude’, as described in the ‘Whisker tracking’ paragraph in the Methods section. In brief, monitoring of whisker movements was performed at high temporal resolution (500 Hz, which is sufficiently high to prevent aliasing). We then calculated the envelope of whisking amplitude as the difference in maximum and minimum whisker angle along a sliding window equal to the imaging frame duration (142 ms). All subsequent analysis was based on this whisking envelope amplitude. We realize that our use of ‘whisking amplitude’ may have caused confusion and was insufficiently introduced. We therefore took the advice of the reviewers and clarified the limitations in temporal resolution and the use of ‘whisking amplitude’ in several places in the manuscript:

1) We extended the sentences at the end of the Introduction and included the suggested reference:

”…Expanding on our recent work on the activation of divergent projection pathways Expanding our recent work on the activity of divergent projection pathways originating in S1 during a texture discrimination task (Chen et al., 2013b; Chen et al., 2015), we sought here to examine how population activity in S1 and S2 evolves over time during such tactile whisker-based behavior. […] Our main goal was, however, to take advantage of the ability to simultaneously image in S1 and S2 and to investigate how the subsets of reciprocally projecting neurons contribute to the coordination of activity across these areas and to the coding of sensory and behavior information.”

2) We clarified the measurements and analysis of whisking in the Results section:

“Whisker movements were monitored with high-speed videography (500 Hz) and licking behavior was measured with a piezo film attached to the water spout. Whisking and licking recordings were downsampled to match the frame rate of imaging (7 Hz), allowing analysis of how neuronal activity relates to slow amplitude changes of whisking envelope and to the occurrence of whisker-texture touches (Materials and methods).”

3) We now explicitly use the term ‘whisking envelope amplitude’ in the legends to Figure 3 and Figure 3—figure supplement 1.

These additions and modifications hopefully now clarify the limits set by our two-photon imaging frame rate and how we dealt with it.

The key part of the paper is the correlation of activity across S1 and S2 as a function of behavioral response. Because the sampling is so slow, this is performed in terms of a population response. The reviewers felt that the linear discriminant analysis was not well explained. Please describe the algorithm used, preferably including the equations on which this was based. For example, was Fisher's linear discriminant derived from the centroids of the responses (with each time point as a dimension) for the hit and correct rejection trials?

We apologize for having failed to explain the linear discriminant analysis. We now provide a detailed mathematical description of the LDA in the Methods section (subsection “Linear discriminant analysis”), explicitly defining the variables used in the analysis. Note that LDA was performed at each time point for the population vectors in state space (with all active neurons in an imaging area as dimensions), thus effectively providing a one-dimensional description of population dynamics with respect to two chosen conditions *C*_1_ and *C*_2_ (e.g., Hit vs. CR or low- vs. high-amplitude whisking; see [Supplementary-material SD1-data]). We also better explain the subsequent analysis of the inter-areal correlation (LDCC_S1:S2_) and the shuffling approach to test the specific contribution of identified mutually projecting neurons (ΔLDCC_S1:S2_).

Is Figure 5 the only aspect of the paper that truly takes advantage of simultaneous imaging of neurons in both regions? The authors should acknowledge this limitation of their experimental design (namely, that in order to maximize the number of neurons imaged through plane hopping, only 10% of the trials allowed them to analyze recordings from the same cohorts of neurons in both regions).

Figure 5 and Figure 6 make use of simultaneous imaging by analyzing the inter-areal correlation LDCC_S1:S2_. Our 3x3 plane hopping implementation provides us with 9 combinations of planes that can be analyzed. The reviewers are correct that this effectively enabled us to utilize 1/9^th^ (11%) of all trials for each animal for cross-areal analysis. However, since animals were imaged for at least 1800 trials this still provided us a sample size of ~200 trials for each combination of populations, which was sufficient for analysis of cross-areal correlations. Importantly, compared to measurements without plane hopping, 9 times more combinations of populations could be used for analysis this way, which was our intention as mentioned in the Results section. We have tried to better clarify this trade-off between number of simultaneously imaged populations and number of trials usable for analysis in the Results section (subsection “Imaging anatomically-identified projection neurons across S1 and S2**”**).

There are other areas where the paper is difficult to follow, which could limit its potential impact. We therefore recommend that you go over the text carefully, to ensure that readers don't have to guess what was done. In particular, it is essential to adequately explain the technical aspects of the various analyses used for the general audience that eLife seeks to reach.

We understand the concern of the reviewers and carefully went through the manuscript again to identify difficult text passages and insufficient descriptions of technical aspects of our analysis methods. Besides amending the description of LDA (see above) we have made several textual adaptations in the main text, especially in the Results section. We hope these modifications have helped to make the manuscript better understandable for a general audience.

Although the reviewers are not asking for new experiments to be carried out, can you comment on whether the discriminative power of S2S1 neurons changes as the mice learn the task?

This is definitely an interesting question but we did not image S2_S1_ neurons (or any other neurons) during task learning in this study. In our previous study (Chen et al. 2015), we did image S1_S2_ neurons in barrel cortex during learning and observed that decision-related activity emerged in these neurons. Given the recent findings from the O’Connor lab that decision-related activity in S1 is inherited from S2 (Yang et al., Nat Neuro 2016), we speculate that S2_S1_ neurons might follow similar changes during learning as S1_S2_ neurons. We are convinced that the required experiments are feasible and we think this will be an interesting future study.

Your proposal that S2 projection neurons provide S1 with specific contextual information was felt to be too speculative in the absence of some manipulation showing that disrupting the activity of these cells affects task performance. The Discussion should be rephrased accordingly.

We have rephrase this, but in particular also referred to further recent evidence from Yang et al. (2016) supporting the notion that S1 inherits some behavior-related activity from S2. (see Discussion, third paragraph).

Finally, the reviewers felt that greater acknowledgement should have been made to similar systems that have been developed in several other labs (Lecoq et al., 2014, Nature Neurosci; Stirman et al., 2014, bioRxiv; Tsai et al., 2015). In particular, the study by Stirman et al. is not cited.

We have made greater acknowledgment of these systems (see Introduction, second paragraph, and Discussion, fourth paragraph).